# Deep Goal-Oriented Clustering

## Abstract

Clustering and prediction are two primary tasks in the fields of unsupervised and supervised machine learning. Although much of the recent advances in machine learning have been centered around those two tasks, the interdependent, mutually beneficial relationship between them is rarely explored. In this work, we hypothesize that a better prediction performance for the downstream task would inform a more appropriate clustering strategy. To this end, we introduce Deep Goal-Oriented Clustering (`DGC`), a probabilistic framework built upon a variational autoencoder with the latent prior being a Gaussian mixture distribution. `DGC` clusters the data by jointly predicting the *side-information* and modeling the inherent data structure in an end-to-end fashion. We show the effectiveness of our model on a range of datasets by achieving good prediction accuracies on the side-information, while, more importantly in our setting, simultaneously learning congruent clustering strategies that are on par with the state-of-the-art. We also apply `DGC` to a real-world breast cancer dataset and show that the discovered clusters carry clinical significance.

## 1 Introduction

Many of the advances in supervised learning in the past decade are due to the development of deep neural networks (DNN), a class of hierarchical function approximators that are capable of learning complex input-output relationships. Prime examples of such advances include image recognition (Krizhevsky et al., 2012), speech recognition (Nassif et al., 2019), and neural translation (Bahdanau et al., 2015). However, with the explosion of the size of modern datasets, it becomes increasingly unrealistic to manually annotate all available data for training (Russakovsky et al., 2015). Therefore, understanding inherent data structure through unsupervised clustering is of increasing importance.

Applying DNNs to unsupervised clustering has been studied in the past few years (Caron et al., 2018; Law et al., 2017; Shaham et al., 2018; Tsai et al., 2021), centering around the concept that the input space in which traditional clustering algorithms operate is of importance. Hence, learning this space from data is desirable. Despite the improvements these approaches have made on benchmark clustering datasets, the ill-defined, ambiguous nature of clustering remains a challenge. Such ambiguity is particularly problematic in scientific discovery, sometimes requiring researchers to choose from different, but potentially equally meaningful clustering results when little information is available a priori (Ronan et al., 2016).

When facing such ambiguity, using side-information to reduce clustering ambivalence proves to be a fruitful direction (Xing et al., 2002; Khashabi et al., 2015; Jin et al., 2013). In general, side-information can be categorized as direct or indirect with respect to the final clustering task. Direct side-information straightforwardly details how the data samples should be clustered, and is usually available in terms of constraints, such as the *must-link* and the *cannot-link* constraints (Wang & Davidson, 2010; Wagstaff & Cardie, 2000), or via a pre-conceived notion of similarity (Xing et al., 2002). However, such direct information requires advanced prior knowledge about the clustering task and intensive manual labeling, thus it is rarely available in reality a priori. Consequently, we focus on indirect side-information, which we define as information that carries useful signals on how the clusters should be formed, but its relation to the clustering task needs to be *learned* and cannot be directly utilized. For instance, if the task is to cluster a group of patients, the side-information could be a medical test with continuous-valued outcome. In this case, we want to use the regression task for predicting the test outcomes to aid the clustering process. To this end, we design a

framework that learns from such indirect side-information and incorporates the learned knowledge into the final clustering result.

**Main Contributions**   We propose *Deep Goal-Oriented Clustering* (`DGC`) to incorporate indirect side-information when forming a pertinent clustering strategy. Specifically: 1) We combine supervision via side-information and unsupervised data structure modeling in a probabilistic manner; 2) We make minimal assumptions on what form the supervised side-information might take, and assume no explicit correspondence between the side-information and the clusters; 3) We train `DGC` end-to-end so that the model simultaneously learns from the available side-information while forming a desirable clustering strategy.

## 2   Related Work

We divide related works in the literature into two categories: 1) methods that utilize direct side-information to form better, less ambiguous clusters (e.g., pairwise constraints); 2) methods that learn from provided labels to lessen the ambiguity in the formed clusters, but rely on the *cluster assumption* (detailed below), and usually assume that the provided discrete labels are the *ground truth labels*. Both classes of methods exclude the possibility of learning from indirectly related, but still informative side-information. Further discussions on the difference between `DGC` and semi-supervised clustering methods can be found in Sec. B in the Appendix.

**Side-information for clustering**   Using pairwise constraints or similarities as side-information to form better clusters has been studied. Wagstaff & Cardie (2000) considered both must-link and cannot-link constraints in the context of K-means clustering. Motivated by image segmentation, Orbanz & Buhmann (2007) proposed a probabilistic model that can incorporate must-link constraints. Khashabi et al. (2015) proposed a nonparametric Bayesian hierarchical model to incorporate pairwise cluster constraints. Vu et al. (2019) utilized constraints and cluster labels as side information. Mazumdar & Saha (2017) gave complexity bounds when provided with an oracle that can be queried for pairwise constraints. Wasid & Ali (2019) incorporated pairwise similarities through the use of fuzzy sets. Manduchi et al. (2021a) incorporated constraint clustering using a deep Gaussian mixture model and optimized the model using stochastic gradient variational inference. Zhang et al. (2021) provided a review of utilizing deep learning frameworks for constraint clustering. Although the term "side information" is used in this review, it is refered to constraints as pairwise or triplet constraints. In supervised clustering, the side-information is the a priori known complete clustering for the training set, which is then being used as a constraint to learn a mapping between the data and the given clustering (Finley & Joachims, 2005). In contrast, importantly, we do not assume any known constraints a priori. Instead, we let the model *learn* what it deems useful from the side-information to guide the clustering process. Therefore, the goal of this work is to leverage informative side-information that is *not* in the form of constraints for learning better clustering strategies.

**The *cluster assumption***   If there exists a set of semantic labels associated with the data (e.g. the digit information for MNIST images), the *cluster assumption* states that the decision boundaries for the semantic labels should not cross high density regions, but instead lie in low density regions (Färber et al., 2010; Chapelle et al., 2006). As a concrete example, Kingma et al. (2014) introduced a hierarchical generative model with two variational layers. Originally meant for semi-supervised classification tasks, it can also be used for clustering, in which case all labels are treated as missing since they are the cluster indices. This implies it has to strictly rely on the *cluster assumption*. We show that this approach is a special case of our framework without the probabilistic ensemble component (see Sec. 4.2). The cluster assumption is restrictive, especially in the case of utilizing indirect side-information where the information is informative but does not directly correspond to the clusters. There exist a line of non-deep learning approaches that attempted to relax the cluster assumption (Varol et al., 2017; Joulin et al., 2010; DeSantis et al., 2012; Chapfuwa et al., 2020). Bair (2013) systematically reviewed semi-supervised clustering methods that find clusters associated with an outcome variable. For instance, Sansone et al. (2016) proposed to model the cluster indices and the class labels separately, underscoring the possibility that each cluster may consist of multiple class labels. Nevertheless, unlike `DGC`, their approach cannot make use of continuous side-information. More importantly, the aforementioned approaches cannot be easily scaled to large, high-dimensional datasets, motivating us to

develop a probabilistic, deep network-based approach that does not rely on the cluster assumption and can be applied to modern benchmark datasets.

**Joint modeling** Previous works on joint modeling/latent space sharing between unsupervised and supervised signals exist. Blei & McAuliffe (2007) incorporated supervision into the latent Dirichlet allocation (LDA) model for document classification. Le et al. (2018) showed that an autoencoder that jointly predicts the targets and the inputs improves performance. Xie & Ma (2019) jointly modeled the reconstruction of a sentence pair and the prediction of the pair's similarity in a `VAE` (Kingma & Welling, 2014) framework. Nagpal et al. (2020) jointly learns deep nonlinear representations to estimate relative risks in time-to-event prediction problems through a mixture modeling approach. We extend the joint modeling literature to clustering and to challenge the commonly assumed cluster assumption. Most similar to our work, Manduchi et al. (2021b) developed a joint modeling framework based on `VAE` with a Gaussian mixture prior. Besides the supervised signals they utilized are survival data, the most critical difference between their work and `DGC` is that how $q(c|\mathbf{y}, \mathbf{z})$ (or in their case $q(c|\mathbf{t}, \mathbf{z})$) is computed. In Manduchi et al. (2021b), they simply chose $q(c|\mathbf{t}, \mathbf{z})$ to be the unsupervised probability $p(c|\mathbf{z}, \mathbf{t})$, the same as what was being done in the original unsupervised VaDE model (Jiang et al., 2017). As shown in Jiang et al. (2017), simply choosing the variation probability $q(c|\mathbf{x})$ in VaDE to be $p(c|\mathbf{x})$ maximizes the ELBO. However, as we show in Sec. 4.3, the same choice, as it was being done in Manduchi et al. (2021b), is sub-optimal in the presence of side-information. In comparison, `DGC`we analytically derived the optimal solution for $q(c|\mathbf{y}, \mathbf{z})$ in terms of maximizing the ELBO (see Sec. 4.3). We showed in Sec 5.2, specifically Tab. 1, that it is difficult for neural networks to recover this analytically derived optimal solution.

## 3 Background & Problem Setup

### 3.1 Background—Variational Deep Embedding

The backbone of `DGC` is the *variational auto-encoder* (VAE) (Kingma & Welling, 2014) with the prior distribution of the latent code being a Gaussian mixture distribution, introduced in Jiang et al. (2017) as VaDE. We next briefly review VaDE. We adopt the notation that lower case letters denote samples from their corresponding distributions; bold, lower case letters denote random variables/vectors; and bold upper case letters denote random matrices.

VaDE assumes the prior distribution of the latent code, $\mathbf{z}$, belongs to the family of Gaussian mixture distributions, i.e., $p(\mathbf{z}) = \sum_c p(\mathbf{z}|c)p(c) = \sum_c \pi_c \mathcal{N}(\mu_c, \sigma_c^2 \mathbf{I})$ where $c$ is a random variable indexing the mixture component $p(\mathbf{z}|c)$ that is assumed to be a normal distribution with mean $\mu_c$ and variance $\sigma_c^2$. The prior probability of each component is assumed to be $\pi_c$. VaDE allows for the clustering of the input data in the latent space, with each component of the Gaussian mixture prior representing a cluster. Since a VAE-based model can be efficiently summarized in terms of its generative and inference processes, we describe VaDE from this perspective. Given an input $\mathbf{x} \in R^d$, the following decomposition of the joint probability $p(\mathbf{x}, \mathbf{z}, c)$ details VaDE's generative process: $p(\mathbf{x}, \mathbf{z}, c) = p(\mathbf{x}|\mathbf{z})p(\mathbf{z}|c)p(c)$. In words, we sample the component index $c$ from a prior categorical distribution $p(c)$, sample the latent code $\mathbf{z}$ from the component $p(\mathbf{z}|c)$, and lastly reconstruct the input $\mathbf{x}$ through the reconstruction network $p(\mathbf{x}|\mathbf{z})$. To perform inference, VaDE is constructed to maximize the log-likelihood of the input data $\mathbf{x}$ by maximizing its *evidence lower bound* (ELBO). With certain assumptions on the prior and variational posterior distributions, the ELBO admits a closed-form expression in terms of the parameters of those distributions. We refer readers to Jiang et al. (2017) for additional details.

### 3.2 Problem Setup

We denote the side-information as $\mathbf{y}$, where $\mathbf{y}$ can be discrete labels or continuous values. Our goal is to leverage $\mathbf{y}$ to inform a better clustering strategy. Abstractly, given the input, side-information random variable pair $(\mathbf{x}, \mathbf{y})$, we seek to divide the input space of $\mathbf{x}$ into non-overlapping subspaces that are meaningful in explaining $\mathbf{y}$. In other words, given a sampled dataset, $\{x_i, y_i\}_{i=1}^n$ of size $n$, we use the prediction task of predicting the side-information $y_i$ using the corresponding input $x_i$ as a *teaching agent* to guide the process

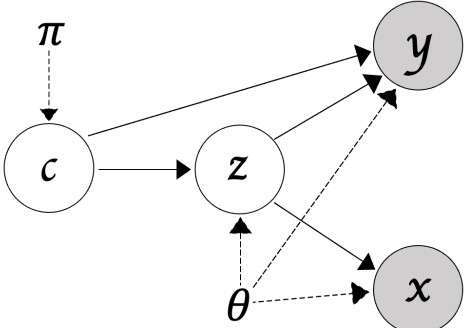

Figure 1: The Bayesian network that underlies the generative process of `DGC`. $\theta$ and $\pi$ together constitute the generative parameters. Solid lines indicate dependencies among the random variables whereas dashed lines indicate dependencies between the random variables and the generative parameters.

of grouping the input set $\{x_i\}_i$ into clusters that optimally explain $\{y_i\}_i$. Since our goal is to discover the optimal subspace-structure, or clusters, without knowing a priori if such a structure indeed exists, a probabilistic framework is more appropriate due to its ability to reason with uncertainty. To this end, we extend the VaDE framework to incorporate the side-information $\mathbf{y}$. Foundationally, we assume $\mathbf{x}$ and $\mathbf{y}$ are correlated, i.e. the input $\mathbf{x}$ carries predictive information with respect to the side-information $\mathbf{y}$. Since the latent code $\mathbf{z}$ is optimized to inherit sufficient information from which the input $\mathbf{x}$ can be reconstructed, it is reasonable to assume that $\mathbf{z}$ also inherits that predictive information. This implies that $\mathbf{x}$ and $\mathbf{y}$ are conditionally independent given $\mathbf{z}$, i.e., $p(\mathbf{x}, \mathbf{y}|\mathbf{z}) = p(\mathbf{x}|\mathbf{z})p(\mathbf{y}|\mathbf{z})$.

## 4 Deep Goal-Oriented Clustering

### 4.1 Generative Process

`DGC` clusters the input $\mathbf{x}$ by clustering its latent representation $\mathbf{z}$. As motivated in Sec. 3.2, we assume that $\mathbf{y}$ manifests differently with respect to the different clusters of $\mathbf{z}$, and by extension, $\mathbf{x}$. This is to say, when learning a predictive functional mapping from $\mathbf{z}$ to $\mathbf{y}$, we assume that the ground truth transformation function, $g_c$, is different for each cluster indexed by $c$. As a result, we learn a different mapping function for each cluster. The overall generative process of our model is as follows: **1**. Generate $c \sim \mathrm{Cat}(\pi)$; **2**. Generate $z \sim p(\mathbf{z}|c)$; **3**. Generate $x \sim p(\mathbf{x}|\mathbf{z})$; **4**. Generate $y \sim p(\mathbf{y}|\mathbf{z}, c)$. The Bayesian network that underlies `DGC` is shown in Fig. 1, and the joint distribution of $\mathbf{x}, \mathbf{y}, \mathbf{z}$, and $c$ can be decomposed as: $p(\mathbf{x}, \mathbf{y}, \mathbf{z}, c) = p(\mathbf{y}|\mathbf{z}, c)p(\mathbf{x}|\mathbf{z})p(\mathbf{z}|c)p(c)$.

### 4.2 Variational Lower Bound & Inference

We learn a variational distribution over the the latent code $\mathbf{z}$ and the cluster index $c$. The joint variational posterior distribution $q(\mathbf{z}, c|\mathbf{x}, \mathbf{y})$ can be factorized as $q(\mathbf{z}, c|\mathbf{x}, \mathbf{y}) = q(\mathbf{z}|\mathbf{x}, \mathbf{y}) \cdot q(c|\mathbf{z}, \mathbf{x}, \mathbf{y})$. Inspired by the usual regression and autoencoding setups where $\mathbf{x}$ is the sole input, we design `DGC` to distill information from $\mathbf{x}$ alone to learn latent representations that are aware of both tasks. Therefore, we use $\mathbf{x}$ as the *sole* input to the encoder that maps $\mathbf{x}$ to $\mathbf{z}$ and omit the variable $\mathbf{y}$ in $q(\mathbf{z}|\mathbf{x}, \mathbf{y})$ for the rest of this work. We demonstrate the advantage of this design over using both $\mathbf{x}$ and $\mathbf{y}$ as input (i.e. the concatenation approach) in Sec. 5.3. With this setup, we have the following variational lower bound (see Sec. A in the Appendix for a detailed derivation)

$$
\begin{aligned}
\log p(\mathbf{x}, \mathbf{y}) \geq\ & \mathbb{E}_{q(\mathbf{z}, c|\mathbf{x}, \mathbf{y})} \log p(\mathbf{y}|\mathbf{z}, c) \\
& + \mathbb{E}_{q(\mathbf{z}, c|\mathbf{x}, \mathbf{y})} \log \frac{p(\mathbf{x}, \mathbf{z}, c)}{q(\mathbf{z}, c|\mathbf{x}, \mathbf{y})} = \mathcal{L}_{\mathrm{ELBO}} \,.
\end{aligned}
\tag{1}
$$

The first term in $\mathcal{L}_{\text{ELBO}}$ allows for a probabilistic ensemble of predictors for $\mathbf{y}$ based on the cluster index. This can be seen as follows

$$
\begin{aligned}
\mathbb{E}_{q(\mathbf{z},c|\mathbf{x},\mathbf{y})} \log p(\mathbf{y}|\mathbf{z},c) &= \mathbb{E}_{q(\mathbf{z}|\mathbf{x})} \left[ \mathbb{E}_{q(c|\mathbf{x},\mathbf{z},\mathbf{y})} \log p(\mathbf{y}|\mathbf{z},c) \right] \\
&= \mathbb{E}_{q(\mathbf{z}|\mathbf{x})} \left[ \sum_{c'} \lambda_{c'} \log p(\mathbf{y}|\mathbf{z},c') \right] \\
&\approx \frac{1}{M} \sum_{l=1}^{M} \left[ \sum_{c'} \lambda_{c'} \log p(\mathbf{y}|\mathbf{z}^{(l)},c') \right]
\end{aligned}
\tag{2}
$$

where $\lambda_{c'} = q(c = c'|\mathbf{x}, \mathbf{z}, \mathbf{y})$ and $l$ indexes the Monte Carlo samples used to approximate the expectation with respect to $q(\mathbf{z}|\mathbf{x})$. The probabilistic ensemble allows the model to maintain necessary uncertainty until an unambiguous clustering structure is captured.

As a side note, the variational lower bound described in Eq. 1 holds regardless of the prior distribution we choose for $\mathbf{z}$. Although we choose the mixture distribution as the prior in this work, choosing $\mathbf{z} \sim \mathcal{N}(0, \mathbf{I})$ and disregarding the probabilistic ensemble component would recover the exact model introduced in Kingma et al. (2014) (when all labels are missing), which is therefore a special case of `DGC`.

### 4.3 Mean-field Variational Posterior Distributions

Following VAE (Kingma et al., 2014), we choose $q(\mathbf{z}|\mathbf{x})$ to be $\mathcal{N}\left(\mathbf{z}|\tilde{\mu}_{\mathbf{z}}, \tilde{\sigma}_{\mathbf{z}}^2 I\right)$ where $\left[\tilde{\mu}_{\mathbf{z}}, \tilde{\sigma}_{\mathbf{z}}^2\right] = h_\theta(\mathbf{x}; \theta)$. $h_\theta$ is parameterized by a feed-forward neural network with weights $\theta$. Although it may seem unnatural to use a unimodal distribution to approximate a multimodal distribution, when the learned $q(c|\mathbf{x}, \mathbf{z}, \mathbf{y})$ becomes discriminative, dissecting the $\mathcal{L}_{\text{ELBO}}$ in the following way indicates such an approximation will not incur a sizeable information loss (see Sec. A in the Appendix for derivation):

$$
\begin{aligned}
\mathcal{L}_{\text{ELBO}} = {} & \mathbb{E}_{q(\mathbf{z},c|\mathbf{x},\mathbf{y})} \log p(\mathbf{y}|\mathbf{z},c) + \mathbb{E}_{q(\mathbf{z}|\mathbf{x})} \log p(\mathbf{x}|\mathbf{z}) \\
& - \mathbb{KL}\left(q(c|\mathbf{x},\mathbf{z},\mathbf{y})\|p(c)\right) - \sum_{c'} \lambda_{c'} \mathbb{KL}\left(q(\mathbf{z}|\mathbf{x})\|p(\mathbf{z}|c')\right)
\end{aligned}
\tag{3}
$$

where $\lambda_{c'}$ denotes $q(c = c'|\mathbf{x}, \mathbf{z}, \mathbf{y})$. Analyzing the last term in Eq. equation 3, we notice that if the learned variational posterior $q(c|\mathbf{x}, \mathbf{z}, \mathbf{y})$ is very discriminative and puts most of its weight on one specific index $c$, all but one $\mathbb{KL}$ terms in the weighted sum will be close to zero. Therefore, choosing $q(\mathbf{z}|\mathbf{x})$ to be unimodal to minimize that specific $\mathbb{KL}$ term is appropriate, as $p(\mathbf{z}|c)$ is a unimodal normal distribution for all $c$.

Choosing $q(c|\mathbf{x}, \mathbf{z}, \mathbf{y})$ appropriately requires us to analyze the proposed $\mathcal{L}_{\text{ELBO}}$ in greater detail (see Sec. A in the Appendix for a detailed derivation):

$$
\begin{aligned}
\mathcal{L}_{\text{ELBO}} = {} & \underbrace{\mathbb{E}_{q(\mathbf{z},c|\mathbf{x},\mathbf{y})} \log p(\mathbf{y}|\mathbf{z},c)}_{\textcircled{1}} + \underbrace{\mathbb{E}_{q(\mathbf{z}|\mathbf{x})} \log \frac{p(\mathbf{x},\mathbf{z})}{q(\mathbf{z}|\mathbf{x})}}_{\textcircled{2}} \\
& - \underbrace{\mathbb{E}_{q(\mathbf{z}|\mathbf{x})} \mathbb{KL}\left(q(c|\mathbf{x},\mathbf{z},\mathbf{y})\|p(c|\mathbf{z})\right)}_{\textcircled{3}}.
\end{aligned}
\tag{4}
$$

We make two observations: 1) $\textcircled{2}$ does not depend on $c$; and 2) the expectation over $q(\mathbf{z}|\mathbf{x})$ does not depend on $c$, and thus has no influence over our choice of $q(c|\mathbf{x}, \mathbf{z}, \mathbf{y})$. Therefore, we choose $q(c|\mathbf{x}, \mathbf{z}, \mathbf{y})$ to maximize $(\textcircled{1} - \textcircled{3})$ and ignore the expectation over $q(\mathbf{z}|\mathbf{x})$. Casting finding $q(c|\mathbf{x}, \mathbf{z}, \mathbf{y})$ as an optimization problem, we have

$$
\begin{aligned}
\min_{q(c|\mathbf{x},\mathbf{z},\mathbf{y})} \quad & f_0(q) = \mathbb{KL}\left(q(c|\mathbf{x},\mathbf{z},\mathbf{y})\|p(c|\mathbf{z})\right) \\
& \quad - \mathbb{E}_{q(c|\mathbf{x},\mathbf{z},\mathbf{y})} \log p(\mathbf{y}|\mathbf{z},c) \\
\text{s.t.} \quad & \sum_c q(c|\mathbf{x},\mathbf{z},\mathbf{y}) = 1, \ q(c|\mathbf{x},\mathbf{z},\mathbf{y}) \geq 0, \ \forall c
\end{aligned}
\tag{5}
$$

The objective functional $f_0$ is convex over the probability space of $q$, as the *Kullback-Leibler divergence* is convex in $q$ and the expectation is linear in $q$. Analytically solving the convex program (5) (see Sec. A in the Appendix for a detailed derivation), we obtain

$$q(c = k|\mathbf{x}, \mathbf{z}, \mathbf{y}) = \frac{p(\mathbf{y}|\mathbf{z}, c = k) \cdot p(c = k|\mathbf{z})}{\sum_k p(\mathbf{y}|\mathbf{z}, c = k) \cdot p(c = k|\mathbf{z})} . \tag{6}$$

To facilitate understanding, we interpret Eq. 6 in two extremes. If $\mathbf{y}$ is evenly distributed across clusters, i.e., the ground truth transformations $g_c$ are the same for all $c$, then $q(c|\mathbf{x}, \mathbf{z}, \mathbf{y}) = p(c = k|\mathbf{z})$, recovering the solution in (Jiang et al., 2017). However, if the side-information is informative while $\mathbf{z}$ itself does not admit a clustering structure ($p(c|\mathbf{z})$ is uniform), the likelihoods $\{p(\mathbf{y}|\mathbf{z}, c = k)\}_k$ will dominate $q$. Therefore, one could interpret any in-between scenario as learning to weight the side-information and the inherent data structure based on how strong their signals are.

Last but not least, one would naturally use the ground truth side-information at both training and test times if it is available. Nevertheless, to make our approach as applicable as possible, we follow the typical regression setup and do not assume access to the side-information at test time, which would prohibit us from evaluating $q(c|x, z, y)$ based on Eq. 6 for a test sample $x$. To remedy this, we pre-train a simple neural network, $f$, to predict $y$ based on the input sample $x$. At test time, we then use $\tilde{y} = f(x)$ as a surrogate for the ground truth $y$ to evaluate $q(c|x, z, \tilde{y})$ for a test sample $x$.

### 4.4 A Confidence Booster

For a given pair $(x, y)$, we want the variational posterior indicated by Eq. 6 to be confident. In other words, the entropy of the probability distribution, $q(c|x, z, y)$, should be small when appropriate. To encourage this behavior, we add the entropy of the side-information network for a given $y$ across clusters, i.e., $\mathbb{H}\left(\mathbf{norm}\{p(y|z, c = k)\}_{k=1}^K\right)$, where $K$ denotes the number of clusters, $\mathbb{H}$ denotes the entropy operator, and **norm** denotes the softmax function that transitions $\{p(y|z, c = k)\}_{k=1}^K$ into a proper probability distribution, to the ELBO proposed in (4) as a regularization term. We further note that when $y$ is continuous, $p(y|z, c)$ denotes the likelihood of $y$ under the assumed probability distribution, i.e. $p(\cdot|z, c)$. The introduced regularization in this continuous case encourages high likelihood of $y$ only under one cluster (as likelihood by definition is non-negative and a low entropy of normalized $p(y|z, c)$ for a given $y$ across the clusters $c$ translates to $y$ resulting in high likelihood under only one cluster), similarly to the case of discrete $y$. We also note that we can directly regularize the entropy of $q(c|x, z, y)$ or $p(c|z)$, but we find regularizing the side-information network works best. The total loss we optimize is

$$\mathcal{L}_{\text{Loss}} = \mathcal{L}_{\text{ELBO}} - \mathbb{H}\left(\mathbf{norm}\{p(\mathbf{y}|\mathbf{z}, c = k)\}_{k=1}^K\right) \tag{7}$$

Since $\mathbb{H}\left(\mathbf{norm}\{p(\mathbf{y}|\mathbf{z}, c = k)\}_{k=1}^K\right)$ is non-negative, $\mathcal{L}_{\text{Loss}}$ is still a proper lower bound and the convexity of $\mathcal{L}_{\text{ELBO}}$ with respect to $q(c|\mathbf{x}, \mathbf{z}, \mathbf{y})$ is preserved. Moreover, we note that the added regularizer does not depend on $q(c|\mathbf{x}, \mathbf{z}, \mathbf{y})$, hence the solution proposed in Eq. 6 is still the global optimum for $q(c|\mathbf{x}, \mathbf{z}, \mathbf{y})$ with respect to $\mathcal{L}_{\text{Loss}}$ in Eq. 7.

## 5 Experiments

We investigate the efficacy of `DGC` on a range of datasets. We refer readers to Sec. C in the Appendix for the experimental details for the data split, network architectures, the choices of learning rate and optimizer, etc. Moreover, since the number of clusters desired is a hyperparameter in `DGC`, we provide an additional study using the Street View House Number (SVHN) dataset (Netzer et al., 2011) to investigate the impact of this hyperparameter on `DGC` in the Appendix.

**A general note on side-information and the goal of each experiment** We briefly describe the side-information used in, and the goal of, each experiment:

- In Sec. 5.1, the side-information indicates what digit each image belongs to. This experiment tests if `DGC` can use side-information to improve an already well-performing VaDE.

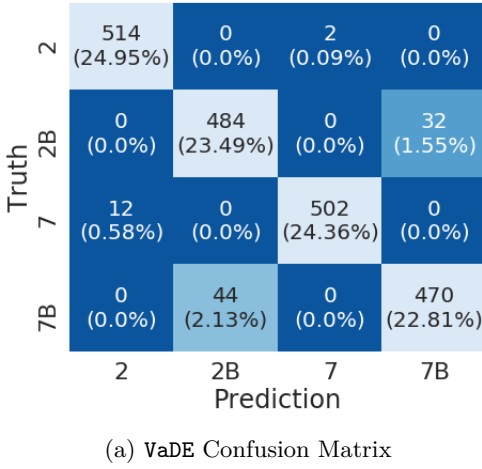
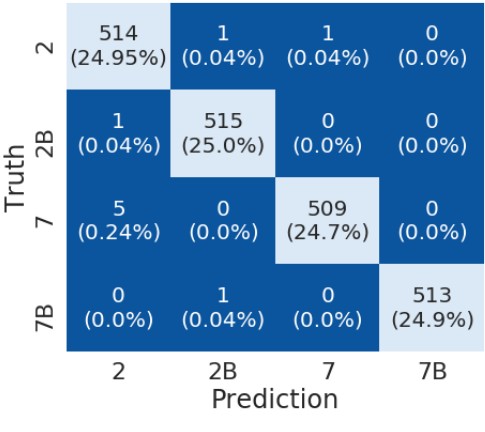

(a) `VaDE` Confusion Matrix          (b) `DGC` Confusion Matrix

Figure 2: Confusion matrices for Noisy MNIST. Abbreviations, 2B/7B, in the row/column labels denotes digits 2/7 with background. Rows represent the predicted clusters, and columns represent the ground truth.

- In Sec. 5.2, the side-information is the continuous value associated with each data point of the Pacman annuli. Our goal is to show that `DGC` can make use of continuous side-information beyond discrete labels.

- In Sec. 5.3, the side-information is the fine-grained labels from the CIFAR 100-20 dataset. This experiment is designed to demonstrate that `DGC` can utilize fine-grained details as side-information to help form meta clusters (i.e. the super-classes) in a natural manner.

- In Sec. 5.4, the side-information is the binary cancer recurrence outcome. The goal of this experiment is to show that `DGC` can discover clusters that unveil patient characteristics that go beyond the recurrence information in a real-world application.

### 5.1 Noisy MNIST

We introduce a synthetic data experiment using the MNIST dataset, which we name the *noisy MNIST*, to illustrate that the supervised part of `DGC` can enhance the performance of an otherwise well-performing unsupervised counterpart. We extract images that correspond to the digits 2 and 7 from MNIST. For each digit, we randomly select half of the images and superpose CIFAR-10 images onto those images as noisy backgrounds (see the Appendix for image samples). The binary side-information $\mathbf{y}$ indicates what digit each image belongs to. Our goal is to identify both digit and background, i.e., to cluster the images into 4 clusters: digits 2 and 7, with and without background. We parameterize the task networks, $\{p(\mathbf{y}|\mathbf{z}, c = k)\}_{k=1}^{4}$, as Bernoulli distributions where we learn the probabilities.

VaDE already performs, achieving a clustering accuracy of 95.6% when the number of clusters is set to 4. Fig. 2a shows that VaDE distinguishes well between the regular and noisy backgrounds, and the incorrectly clustered samples are mainly due to its inability to differentiate the underlying digits. This is reasonable: if the background signal dominates, the network may focus on the background for clustering as it has no explicit knowledge about the digits.

`DGC` performs nearly perfectly with the added side-information, obtaining a clustering accuracy of 99.55%. `DGC` handles the difficulty of distinguishing between digits under the presence of strong, noisy backgrounds well as it makes almost no mistakes in doing so (Fig. 2b). Importantly, the side-information does not overshadow the original advantage of VaDE (i.e., distinguishing whether the images contain background or not). Instead, it enhances the overall model in cases where VaDE struggles. Furthermore, as detailed in Sansone et al. (2016) and earlier sections, most existing approaches that take advantage of available labels rely on *the cluster assumption*, which assumes a one-to-one correspondence between the clusters and the labels used for supervision. This experiment is a concrete example that demonstrates `DGC` does not need to rely on such

**Truth**

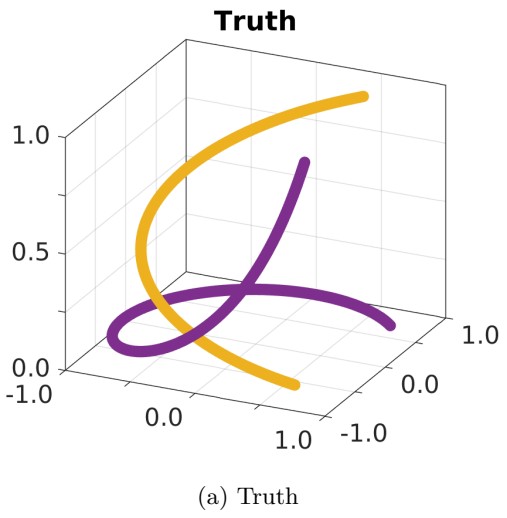

**Samples**

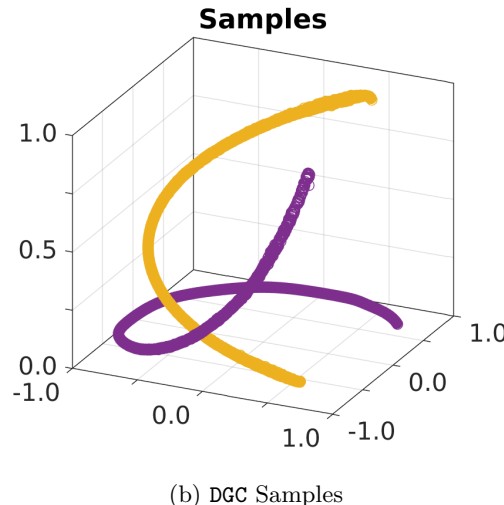

(a) Truth

(b) `DGC` Samples

Figure 3: 3D Pacman truth (a) and `DGC` samples (b).

an assumption to form a sound clustering strategy. Instead, `DGC` is able to work with side-information that is only partially indicative of what the final clustering strategy should be, making `DGC` more applicable to general settings. We also include an ablation study where we ablate the unsupervised component of `DGC`(i.e. VaDE) in the Appendix to show both the supervised and the unsupervised components are needed for good clustering performance.

**Noisy MNIST—Ablation Study**   We further explore the behavior of `DGC` without its unsupervised part to demonstrate the importance of capturing the inherent data structure beyond the side-information. We ablate the probabilistic components (i.e., we get rid of the decoder and the loss terms associated with it, so that only the supervision will inform how the clusters are formed in the latent space) and perform clustering using only the supervised part of our model. We find that clustering accuracy degrades from the nearly-perfect accuracy obtained by the full model to 50%. Coupled with the improvements over VaDE, this indicates that each component of our model contributes to the final accuracy and that our original intuition that side-information and clustering may reinforce each other is correct.

### 5.2   Pacman

In this experiment we demonstrate `DGC`'s ability to utilize *continuous* side-information, i.e. regression tasks, to aid clustering.  We introduce the Pacman dataset, a Pacman-shaped data consisting of two annuli. Furthermore, each point on the two annuli is associated with a continuous value (see the Appendix for a detailed explanation) as the side-information. These values are constructed such that they decrease linearly (from 1 to 0) in one direction for the inner (yellow) annulus, and increase exponentially (from 0 to 1) in the opposite direction for the outer (purple) annulus (see Figure 3a for a 3D illustration of the dataset). We use linear/exponential rates for the side-information to not only test our model's ability to detect different trends, but also to test its ability to fit different rates of change in values.

Our goal is to separate the two annuli depicted in Fig. 3a. This is challenging as the annuli were deliberately chosen to be very close to each other. We first applied various traditional clustering methods, such as the K-means and hierarchical clustering algorithms, to cluster the 2D Pacman-shaped data (i.e., not using the side-information, but only the 2D Cartesian coordinates). Besides hierarchical clustering with single linkage (and not other distance metric), none of the traditional methods managed to separate the two annuli. Moreover, these approaches also produce different clustering results as they are based on different distance metrics (see the Appendix for these results). This phenomenon echos the fundamental problem for unsupervised clustering: the concept of clustering is inherently subjective, and different approaches can potentially produce different, but sometimes equally meaningful, clustering results. Applying `DGC` with the

input $x$ being the 2D Cartesian point coordinates and the side-information $y$ being the aforementioned continuous values, and parameterizing the task networks, $\{p(\mathbf{y}|\mathbf{z}, c = k)\}_{k=1}^{2}$, as Gaussian distributions where we learn the means and the covariance matrices, DGC distinguishes the two annulli wholly based on the discriminative information carried by the side-information. Moreover, since our approach is generative in nature, we can generate samples from the model once it is trained. We show the generated samples in Fig. 3b, where the Pacman shape, the linear trend, and the exponential trend are adequately captured. This corroborates the model's ability to incorporate continuous side-information. This ability is highly attractive, as it lends itself to any general regression setting in which one believes the optimal clustering structure should be informed by the regression task.

Finally, we compare DGC to VaDE, its ablated version, and a baseline method to substantiate the efficacy of our proposed framework. We next describe the ablated version and the baseline method. First, although the solution to the convex program in Eq. 5 provides the global optimal choice of $q(c|\mathbf{x}, \mathbf{z}, \mathbf{y})$ from a theoretical standpoint, our proposed framework, specifically the proposed $\mathcal{L}_{\text{ELBO}}$ (Eq. 1), holds for any choice of $q(c|\mathbf{x}, \mathbf{z}, \mathbf{y})$. We thus ablate the convex program component of our model and parameterize $q(c|\mathbf{x}, \mathbf{z}, \mathbf{y})$ directly

Table 1: Pacman accuracies

| Models | ACC |
|--------|-----|
| VaDE | $50.4\% \pm 0\%$ |
| NN-DGC | $81.6\% \pm 5.3\%$ |
| AUG-SS | $82.3\% \pm 4.6\%$ |
| DGC | **$99.4\% \pm 0.3\%$** |

using a neural network (named as NN-DGC). Second, recall (in Sec. 4.2) by choosing $\mathbf{z} \sim \mathcal{N}(0, \mathbf{I})$, the unsupervised part of DGC recovers exactly the semi-supervised (SS) approach introduced in Kingma et al. (2014) in the case when all labels (that correspond to the clusters in our case) are missing. Since SS approaches are not expected to perform well in a purely unsupervised setting, we include the probabilistic ensemble component as an augmentation (AUG-SS) to make the comparison more fair.

The results in Tab. 1 are obtained from training each model 100 times, and demonstrate that: 1) without the side-information $\mathbf{y}$, VaDE cannot distinguish between the two annulli, demonstrating the importance of using side-information; 2) the convex program (Eq.5) is crucial to the success of DGC.Although technically possible, it is difficult for a neural network to find the same optimal distribution; 3) the choice of the prior on the latent code $\mathbf{z}$ is of importance, and the Gaussian mixture distribution is more suitable for modeling clusters than an isotropic Gaussian.

### 5.3 CIFAR 100-20

We apply DGC to the CIFAR 100-20 dataset where the dataset setup is ideal for demonstrating the advantage of being able to utilize useful side-information for clustering. Two types of labels are provided for each image: one indicating which 100 fine-grained classes and another one indicating which 20 super-classes the image belongs to.

Aligning with the clustering literature, our goal is to cluster the images into the 20 super-classes. Different from other approaches, our framework is able to utilize the fine-grained classes as the side-information to aid clustering. For baseline comparisons, we compare DGC to VaDE and a baseline K-means model. We also compare DGC to SCAN (Gansbeke et al., 2020), RUC (Park et al., 2020), and the current SOTA approach, SPICE (Niu & Wang, 2021). The results are shown in Tab. 2. Last but not least, we include six additional studies where the modeling flexibility of DGC and, more critically, the importance of incorporating the side-information in a principled manner are demonstrated. We next expound upon the results and our findings.

First, by utilizing the fine-grained information, DGC expectedly outperforms VaDE by a significant margin, substantiating the advantage of using informative side-information. Second, to test if it is only the side-information dominating the clustering accuracy, and to demonstrate the importance of seamlessly incorporating

Table 2: CIFAR 100-20 clustering accuracy

| Models | ACC |
|---|---|
| K-means | 50.4% |
| VaDE | 45.2% |
| SCAN | 50.7% |
| RUC | 54.3% |
| SPICE | 58.4% |
| VaDE-Concat | 48.7% |
| SPICE-Concat | 59.1% |
| DGC | **62.7%** |

the side-information alongside the unsupervised modeling, we perform an ablation study where we apply the K-means algorithm on the last hidden layer embeddings of the input data obtained from the pre-trained network that provides `DGC` with the estimated side-information **y** at test time. As shown in Tab. 2, such a simple baseline does surprisingly well, outperforming VaDE and is on par with SCAN. However, `DGC` still outperforms this baseline by a large margin, indicating that it is beneficial to model the inherent data structure beyond the side-information. Third of all, with the aid of the fine-grained information, `DGC` achieves better accuracy than the SOTA unsupervised method, SPICE, and two other leading performance methods. It is crucial to note that the methods we compare `DGC` to do not, and more importantly cannot, utilize side-information in a *principled* manner, substantiating the theme of this work: utilizing informative side-information helps clustering. To drive this point home, we concretely establish the importance of `DGC` being a principled approach in the second additional study. Last but not least, we emphasize again that we assume no access to the side-information at test time because we intend to make `DGC` as applicable as possible. In this experiment, we achieve a clustering accuracy of 67.1% if we were to use the ground-truth fine-grained labels as the side-information at test time. We further substantiate the effectiveness of `DGC` through the following six addtional studies.

**Modeling flexibility**  As `DGC` is not restricted by the cluster assumption, we can use the 20 super-classes as the side-information to help cluster the images into 100 clusters. Compared to the clustering accuracy of 35.1% for VaDE, `DGC` obtains 47.6%, substantiating the utility of `DGC` in a scenario where using less expensive side-information can help categorize data in a more fine-grained manner.

**Compare to simple concatenation strategy**  We now demonstrate the importance of incorporating side-information in a principled manner. It is easy to see that an ad hoc alternative is to simply concatenate the side-information to the input and use the concatenated input for clustering; therefore, we compare `DGC` to this simple baseline using two models, VaDE (as it is the backbone of `DGC`, denoted as VaDE-Concat) and SPICE (as it is the SOTA method on CIFAR 100-20, denoted as SPICE-Concat). The results in Tab. 2 show that incorporating the side-information through concatenation results in marginal performance improvement while still underperforming `DGC`. Considering `DGC` and VaDE-Concat share the same backbone and utilize the same amount of input information, the benefit of our approach is particularly clear given that `DGC` outperforms VaDE-Concat by a wide margin.

**The Enhanced VaDE-Concat**  We note that we can further alter the generative process of VaDE-Concat such that the decoder is cluster-dependent, i.e. $p(\mathbf{x}^*, \mathbf{z}, c) = p(\mathbf{x}^*|\mathbf{z}, c)p(\mathbf{z}|c)p(c)$ where $\mathbf{x}^*$ denotes the concatenation of **x** and **y**. We note that the analytical solution to the variational distribution $q(c|\mathbf{x})$ derived in Jiang et al. (2017) no longer holds, so we use a neural network to parameterize $q(c|\mathbf{x})$ instead. The enhanced VaDE-Concat achieves a clustering accuracy of 54.5%, a noticeable improvement over VaDE-Concat while still lagging `DGC`. This echos the result in Sec. 5.2 in that obtaining an analytically global optimum solution to $q(c|\mathbf{x})$ is crucial. Despite the flexibility of neural nets, they can get stuck in local minima in an unforeseeable way that would degrade the final result.

**Non-informative side-information**  As alluded in the introduction, `DGC` is motivated by the existence of useful side-information, and therefore naturally `DGC`'s performance depends on the quality of the side-

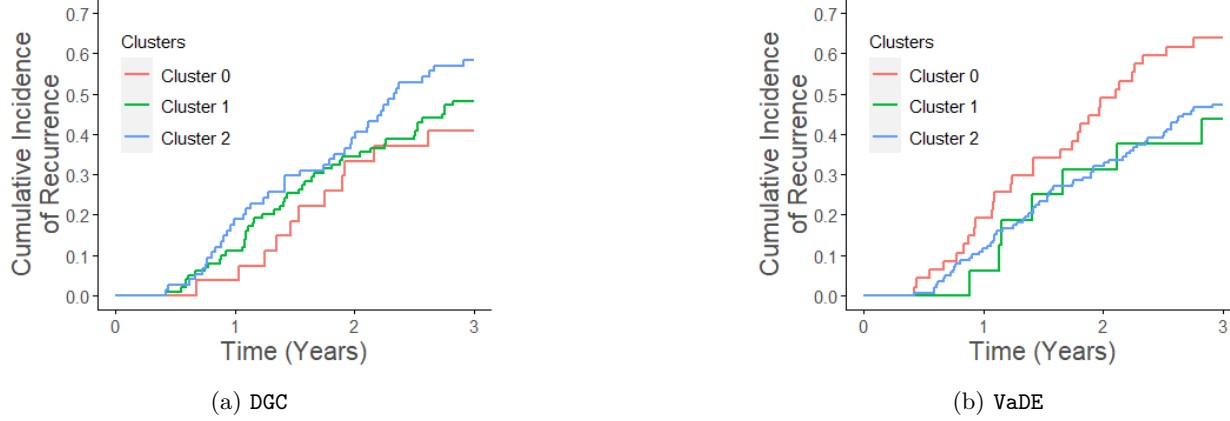

(a) `DGC`  (b) `VaDE`

Figure 4: Kaplan-Meier curves for `DGC` & `VaDE`

information $\mathbf{y}$. Nevertheless, Eq. 6 suggests that `DGC` should resort back to VaDE when $\mathbf{y}$ is not informative. We conduct an experiment where instead of using the fine-grained classes as $\mathbf{y}$, we randomly generate integers between 0 and 99 as $\mathbf{y}$. In this case, we obtain a clustering accuracy of 44.6% (compared to that of 45.2% for VaDE), suggesting that DGC indeed resorts back to VaDE for non-informative side-information.

**Misspecifying the number of clusters**  We study whether `DGC` will be robust to the misspecification of the number of clusters. Instead of setting the number of clusters wanted as 20, we misspecify the number of clusters as 30 and 50, and obtain clustering accuracies of 61.4% and 58.2% (versus the original clustering accuracy of 62.7%). `DGC` obtains a reasonable clustering accuracy even when the number of clusters wanted is set to be more than twice as the expected number, demonstrating the robustness of `DGC` to such misspecification.

**Jointly train the prediction network for the side-information**  In the main manuscript we pre-train a network to predict the side-information $\mathbf{y}$ for test time to make `DGC` as applicable as possible (again, we emphasize that one should use the real $\mathbf{y}$ at both the training and the test times if it is available). Now we jointly train the prediction network along with `DGC` and use the prediction $\hat{\mathbf{y}}$, instead of the real $\mathbf{y}$, at *both* the training and the test times to calculate $q(c|x, \hat{y}, z)$ for a sample $x$. We obtain a clustering accuracy of 61.5%. This is comparable to the clustering accuracy of 62.7% achieved using separate pre-training. Hence, both strategies are viable.

### 5.4  Carolina Breast Cancer Study (CBCS)

We apply `DGC` to a real-world breast cancer dataset collected as part of the Carolina Breast Cancer Study (CBCS). The dataset consists of 1,713 patients, each of which has 2-4 associated histopathological images and a list of biological markers such as the Pam50 gene expressions (Troester et al., 2018) and the estrogen-receptor (ER) status.

We use the binary indicator for breast cancer recurrence as the side-information $\mathbf{y}$. Applying deep learning techniques, supervised or unsupervised, to analyze histopathological images of breast cancer has gained traction in recent years (Xie et al., 2019). Distinguished from those methods, our goal is to inspect whether the discovered clusters, whose formation is influenced both by the recurrence side-information and the unsupervised reconstruction signal, carry meaningful information in terms of survival rate or gene expression beyond the recurrence information. Since `DGC` is not restricted by the cluster assumption, we train three clusters for analysis despite the binary side-information. We parameterize the task networks, $\{p(\mathbf{y}|\mathbf{z}, c = k)\}_{k=1}^3$, as Bernoulli distributions. See the Appendix for experimental details.

To investigate whether the three clusters that we discovered were identifying meaningful differences in tumor biology, we examine the differences in rates of cancer recurrence and features of tumor aggressiveness between the clusters. We also compare to the baseline clusters obtained from the purely unsupervised VaDE to

corroborate the importance of the added side-information. Using a Kaplan-Meier estimator to estimate risk differences for time to cancer recurrence within three years, we obtained a p-value of 0.0024 for the differences in recurrence risk among the clusters obtained by `DGC`, and observed that Cluster 0 had the lowest risk of recurrence (RRD) and Cluster 2 had the highest risk (see Fig. 4a). Furthermore, we observed substantial differences in recurrence risk at three years of follow-up between the clusters, particularly Clusters 0 and 2 (see Fig. 4a). By comparison, with a p-value of 0.073, the differences in recurrence risk among the clusters from VaDE is much less significant than that from `DGC`.

Table 3: `DGC` RRD between clusters

| Comparsion | RRD (95% CI) |
| --- | --- |
| Cluster 1 VS Cluster 2 | 11.3% (-4.4, 26.9) |
| Cluster 0 VS Cluster 2 | 18.8% (-3.1, 40.7) |

In terms of tumor characteristics, cluster that has the highest recurrence rate should have the most negative ER subtype, the highest grade, and the most Basal-like tumor subtype. We observed that for `DGC`, Cluster 0, which has the lowest recurrence rate, contained more indolent tumors, characterized by good-prognosis features such as ER positivity, low grade, and Luminal A tumor subtype (see Tab. 4). In contrast, more aggressive tumor characteristics were featured in Clusters 1 and 2, such as negative ER status, high grade, and Basal-like tumor subtype, although Cluster 1 appeared to be intermediate in some characteristics. Coupled with the differences in cancer outcomes, these differences in tumor characteristics indicate that the method successfully distinguished between tumors with low-risk features (Cluster 0) and tumors with intermediate- and high-risk features (Clusters 1 and 2). We also include the same table that characterizes tumor characteristic for clusters obtained from VaDE in Tab. 5. Contrary to the clusters obtained from `DGC`, Cluster 0 from VaDE, which has the highest recurrence rate (Fig 4b), does not have any of the desired tumor characteristic for high recurrence group, i.e., it does not have the most high grade patients, the most negative ER subtype patients, nor the most Basal-like tumor subtype patients (Cluster 2 has all those characteristics, while being a low recurrence cluster). This corroborates the fact that `DGC` is able to find more meaningful clusters in a real-world application.

## 6 Conclusion

We introduced `DGC`, a probabilistic framework that allows for the integration of both the side-information and the unsupervised information when searching for the optimal clustering structure in the latent space. This is a relevant but daunting task, where previous attempts are either largely restricted to discrete, supervised, ground-truth labels or rely heavily on the side-information being provided as manually tuned pairwise constraints or similarities. To the best of our knowledge, this is the first deep network-based attempt to *learn* from indirect, but informative side-information to find the optimal clustering structure, all the while making minimal assumptions on either the form of the side-information or the relationship between the side-information and the clusters. This method is applicable to a variety of fields where an instance's input and task are defined but its membership, which should be influenced by both, is important and unknown. Training the model in an end-to-end fashion, we demonstrate on various datasets that `DGC` is capable of learning sensible clustering strategies that align with both the side-information and the inherent data structure.

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

Table 4: Tumor characteristics per cluster for `DGC`. Features are color-coded as low , intermediate , or high risk.

| | | Cluster 0 N(%) | Cluster 1 N(%) | Cluster 2 N(%) |
|---|---|---|---|---|
| ER Status | Positive | 20 (74.1) | 58 (58.0) | 43 (57.3) |
| | Negative | 7 (25.9) | 42 (42.0) | 32 (42.7) |
| Grade | Low | 8 (29.6) | 16 (16.0) | 4 (5.3) |
| | Medium | 7 (25.9) | 25 (25.0) | 26 (34.7) |
| | High | 12 (44.4) | 59 (59.0) | 45 (60.0) |
| Tumor Subtype | Luminal A | 14 (51.9) | 28 (28.0) | 17 (22.6) |
| | Luminal B | 7 (25.9) | 20 (20.0) | 18 (24.0) |
| | ER-/HER2+ | 1 (3.7) | 9 (9.0) | 3 (4.0) |
| | Basal-like | 4 (14.8) | 42 (42.0) | 37 (49.3) |

Table 5: Tumor characteristics per cluster for VaDE. Features are color-coded as low , intermediate , or high risk.

| | | Cluster 0 N(%) | Cluster 1 N(%) | Cluster 2 N(%) |
|---|---|---|---|---|
| ER Status | Positive | 24 (51.1) | 9 (56.3) | 88 (63.3) |
| | Negative | 23 (48.9) | 7 (43.8) | 51 (36.7) |
| Grade | Low | 6 (12.8) | 0 (0.0) | 22 (15.8) |
| | Medium | 8 (17.0) | 3 (18.8) | 47 (33.8) |
| | High | 33 (70.2) | 13 (81.2) | 70 (50.4) |
| Tumor Subtype | Luminal A | 8 (22.9) | 3 (23.1) | 44 (44.0) |
| | Luminal B | 5 (14.3) | 3 (23.1) | 16 (16.0) |
| | ER-/HER2+ | 1 (2.9) | 0 (0.0) | 9 (9.0) |
| | Basal-like | 21 (60.0) | 7 (53.8) | 31 (31.0) |

Mathilde Caron, Hugo Touvron, Ishan Misra, Herv'e J'egou, Julien Mairal, Piotr Bojanowski, and Armand Joulin. Emerging properties in self-supervised vision transformers. *ArXiv*, abs/2104.14294, 2021.

Olivier Chapelle, Bernhard Schölkopf, and Alexander Zien. Semi-supervised learning. 2006.

Paidamoyo Chapfuwa, Chunyuan Li, Nikhil Mehta, Lawrence Carin, and Ricardo Henao. Survival cluster analysis. *Proceedings of the ACM Conference on Health, Inference, and Learning*, 2020.

Stacia M DeSantis, E. Andres Houseman, B. Coull, Catherine L. Nutt, and Rebecca A. Betensky. Supervised Bayesian latent class models for high-dimensional data. *Statistics in Medicine*, 31, 2012.

Ines Färber, Stephan Günnemann, Hans-Peter Kriegel, Peer Kröger, Emmanuel Müller, Erich Schubert, Thomas Seidl, Arthur Zimek, and Ludwig-Maximilians-Universitaet Muenchen. On using class-labels in evaluation of clusterings. 2010.

Thomas Finley and Thorsten Joachims. Supervised clustering with support vector machines. In *ICML '05*, 2005.

Wouter Van Gansbeke, Simon Vandenhende, Stamatios Georgoulis, Marc Proesmans, and Luc Van Gool. Scan: Learning to classify images without labels. In *ECCV*, 2020.

Kaiming He, X. Zhang, Shaoqing Ren, and Jian Sun. Deep residual learning for image recognition. *2016 IEEE Conference on Computer Vision and Pattern Recognition (CVPR)*, pp. 770–778, 2016.

Zhuxi Jiang, Yin Zheng, Huachun Tan, Bangsheng Tang, and Hanning Zhou. Variational deep embedding: An unsupervised and generative approach to clustering. In *IJCAI*, 2017.

Xin Jin, Jiebo Luo, Jie Yu, Gang Wang, Dhiraj Joshi, and Jiawei Han. Reinforced similarity integration in image-rich information networks. *IEEE Transactions on Knowledge and Data Engineering*, 25:448–460, 2013.

Armand Joulin, Francis R. Bach, and Jean Ponce. Efficient optimization for discriminative latent class models. In *NIPS*, 2010.

Daniel Khashabi, Jeffrey Yufei Liu, John Wieting, and Feng Liang. Clustering with side information: From a probabilistic model to a deterministic algorithm. *ArXiv*, abs/1508.06235, 2015.

Diederik P. Kingma and Max Welling. Auto-encoding variational Bayes. *CoRR*, abs/1312.6114, 2014.

Diederik P. Kingma, Shakir Mohamed, Danilo Jimenez Rezende, and Max Welling. Semi-supervised learning with deep generative models. In *NIPS*, 2014.

Alex Krizhevsky, Ilya Sutskever, and Geoffrey E. Hinton. Imagenet classification with deep convolutional neural networks. In *NIPS*, 2012.

Marc T. Law, Raquel Urtasun, and Richard S. Zemel. Deep spectral clustering learning. In *ICML*, 2017.

Lei Le, Andrew Patterson, and Martha White. Supervised autoencoders : Improving generalization performance with unsupervised regularizers. 2018.

Laura Manduchi, Kieran Chin-Cheong, Holger Michel, Sven Wellmann, and Julia E. Vogt. Deep conditional gaussian mixture model for constrained clustering. *ArXiv*, abs/2106.06385, 2021a. URL `https://api.semanticscholar.org/CorpusID:235417417`.

Laura Manduchi, Ricards Marcinkevics, Michela Carlotta Massi, Verena Gotta, Timothy Müller, Flavio Vasella, Marian Christoph Neidert, Marc Pfister, and Julia E. Vogt. A deep variational approach to clustering survival data. *ArXiv*, abs/2106.05763, 2021b. URL `https://api.semanticscholar.org/CorpusID:235390687`.

Arya Mazumdar and Barna Saha. Query complexity of clustering with side information, 2017.

Chirag Nagpal, Xinyu Li, and Artur W. Dubrawski. Deep survival machines: Fully parametric survival regression and representation learning for censored data with competing risks. *IEEE Journal of Biomedical and Health Informatics*, 25:3163–3175, 2020. URL `https://api.semanticscholar.org/CorpusID:211817982`.

Ali Bou Nassif, Ismail Shahin, Imtinan B. Attili, Mohammad Azzeh, and Khaled Shaalan. Speech recognition using deep neural networks: A systematic review. *IEEE Access*, 7:19143–19165, 2019.

Yuval Netzer, Tiejie Wang, Adam Coates, Alessandro Bissacco, Baolin Wu, and Andrew Y. Ng. Reading digits in natural images with unsupervised feature learning. 2011.

Chuang Niu and Ge Wang. Spice: Semantic pseudo-labeling for image clustering. *ArXiv*, abs/2103.09382, 2021.

Peter Orbanz and Joachim M. Buhmann. Nonparametric Bayesian image segmentation. *International Journal of Computer Vision*, 77:25–45, 2007.

Sungwon Park, Sungwon Han, Sundong Kim, Danu Kim, Sungkyu Park, Seunghoon Hong, and M. Cha. Improving unsupervised image clustering with robust learning. *ArXiv*, abs/2012.11150, 2020.

Tom Ronan, Zhijie Qi, and Kristen M Naegle. Avoiding common pitfalls when clustering biological data. *Science Signaling*, 9:re6–re6, 2016.

Olga Russakovsky, Jia Deng, Hao Su, Jonathan Krause, Sanjeev Satheesh, Sean Ma, Zhiheng Huang, Andrej Karpathy, Aditya Khosla, Michael S. Bernstein, Alexander C. Berg, and Li Fei-Fei. Imagenet large scale visual recognition challenge. *International Journal of Computer Vision*, 115:211–252, 2015.

Emanuele Sansone, Andrea Passerini, and Francesco Natale. Classtering: Joint classification and clustering with mixture of factor analysers. In *ECAI*, 2016.

Uri Shaham, Kelly Stanton, Haochao Li, Boaz Nadler, Ronen Basri, and Yuval Kluger. Spectralnet: Spectral clustering using deep neural networks. *ArXiv*, abs/1801.01587, 2018.

M. Troester, Xuezheng Sun, Emma H. Allott, J. Geradts, S. Cohen, Chiu-Kit J. Tse, Erin L. Kirk, L. Thorne, M. Mathews, Y. Li, Z. Hu, W. Robinson, K. Hoadley, O. Olopade, K. Reeder-Hayes, H. S. Earp, A. Olshan, L. Carey, and C. Perou. Racial differences in PAM50 subtypes in the carolina breast cancer study. *JNCI: Journal of the National Cancer Institute*, 110, 2018.

Tsung Wei Tsai, Chongxuan Li, and Jun Zhu. Mice: Mixture of contrastive experts for unsupervised image clustering. *ArXiv*, abs/2105.01899, 2021.

E. Varol, Aristeidis Sotiras, and Christos Davatzikos. Hydra: Revealing heterogeneity of imaging and genetic patterns through a multiple max-margin discriminative analysis framework. *NeuroImage*, 145:346–364, 2017.

Viet-Vu Vu, Quan Do, Vu-Tuan Dang, and Do Toan. An efficient density-based clustering with side information and active learning: A case study for facial expression recognition task. *Intelligent Data Analysis*, 23: 227–240, 02 2019.

Kiri Wagstaff and Claire Cardie. Clustering with instance-level constraints. In *AAAI/IAAI*, 2000.

Xiang Wang and Ian Davidson. Flexible constrained spectral clustering. In *KDD '10*, 2010.

Mohammed Wasid and Rashid Ali. Fuzzy side information clustering-based framework for effective recommendations. *Computing and Informatics*, 38:597–620, 01 2019.

Juanying Xie, R. Liu, Joseph Luttrell, and Chaoyang Zhang. Deep learning based analysis of histopathological images of breast cancer. *Frontiers in Genetics*, 10, 2019.

Zhongbin Xie and Shuai Ma. Dual-view variational autoencoders for semi-supervised text matching. In *IJCAI*, 2019.

Eric P. Xing, Andrew Y. Ng, Michael I. Jordan, and Stuart J. Russell. Distance metric learning with application to clustering with side-information. In *NIPS*, 2002.

Linxiao Yang, Ngai-Man Cheung, Jiaying Li, and Jun Fang. Deep clustering by gaussian mixture variational autoencoders with graph embedding. *2019 IEEE/CVF International Conference on Computer Vision (ICCV)*, pp. 6439–6448, 2019.

Hongjing Zhang, Tianyang Zhan, Sugato Basu, and Ian Davidson. A framework for deep constrained clustering. *Data Mining and Knowledge Discovery*, 35:593 – 620, 2021. URL `https://api.semanticscholar.org/CorpusID:231418939`.

## A    Mathematical Details

This section provides detailed derivations for the theoretical claims made in the main manuscript.

**Proposition 1.** *To explain the fact that choosing $q(\boldsymbol{z}|\boldsymbol{x})$ to be unimodal will not incur a sizable information loss when the learned $q(c|\boldsymbol{x})$ is discriminative, we dissect the $\mathcal{L}_{ELBO}$ as follows (Eq.4 in the main paper)*

$$
\begin{aligned}
\mathcal{L}_{ELBO} = {}& \mathbb{E}_{q(z,c|\boldsymbol{x},\boldsymbol{y})} \log p(\boldsymbol{y}|\boldsymbol{z},c) + \mathbb{E}_{q(z|\boldsymbol{x})} \log p(\boldsymbol{x}|\boldsymbol{z}) \\
& - \mathbb{KL}\left( q(c|\boldsymbol{y},\boldsymbol{z}) || p(c) \right) \\
& - \sum_k \lambda_k \mathbb{KL}\left( q(\boldsymbol{z}|\boldsymbol{x}) || p(\boldsymbol{z}|c=k) \right) .
\end{aligned}
\tag{8}
$$

*Proof.* The derivation can be seen as follows

$$
\begin{aligned}
\mathcal{L}_{\text{ELBO}} &= \mathbb{E}_{q(\mathbf{z},c|\mathbf{x},\mathbf{y})} \log p(\mathbf{y}|\mathbf{z},c) \\
&\quad + \mathbb{E}_{q(\mathbf{z},c|\mathbf{x},\mathbf{y})} \log \frac{p(\mathbf{x},\mathbf{z},c)}{q(\mathbf{z},c|\mathbf{x},\mathbf{y})} \\
&= \mathbb{E}_{q(\mathbf{z},c|\mathbf{x},\mathbf{y})} \log p(\mathbf{y}|\mathbf{z},c) \\
&\quad + \mathbb{E}_{q(\mathbf{z},c|\mathbf{x},\mathbf{y})} \log \frac{p(\mathbf{x}|\mathbf{z})p(\mathbf{z}|c)p(c)}{q(\mathbf{z}|\mathbf{x})q(c|\mathbf{z},\mathbf{y})} \\
&= \mathbb{E}_{q(\mathbf{z},c|\mathbf{x},\mathbf{y})} \log p(\mathbf{y}|\mathbf{z},c) + \mathbb{E}_{q(\mathbf{z}|\mathbf{x})} \log p(\mathbf{x}|\mathbf{z}) \\
&\quad - \mathbb{KL}\left(q(c|\mathbf{y},\mathbf{z})||p(c)\right) + \mathbb{E}_{q(\mathbf{z},c|\mathbf{x},\mathbf{y})} \log \frac{p(\mathbf{z}|c)}{q(\mathbf{z}|\mathbf{x})}
\end{aligned}
\tag{9}
$$

where

$$
\begin{aligned}
\mathbb{E}_{q(\mathbf{z},c|\mathbf{x},\mathbf{y})} \log \frac{p(\mathbf{z}|c)}{q(\mathbf{z}|\mathbf{x})} &= \mathbb{E}_{q(c|\mathbf{y},\mathbf{z})}\mathbb{E}_{q(\mathbf{z}|\mathbf{x})} \log \frac{p(\mathbf{z}|c)}{q(\mathbf{z}|\mathbf{x})} \\
&= -\sum_{k} \lambda_k \mathbb{KL}\left(q(\mathbf{z}|\mathbf{x})||p(\mathbf{z}|c=k)\right)
\end{aligned}
\tag{10}
$$

where $\lambda_k = q(c=k|\mathbf{y},\mathbf{z})$. We note that we omit $\mathbf{y}$ in $q(\mathbf{z}|\mathrm{x},\mathbf{y})$ (hence $q(\mathbf{z}|\mathrm{x})$) because the aforementioned reasoning that we want to encourage the encoder to extract information from $\mathbf{x}$ alone that is enough to reconstruct $\mathbf{x}$ and predict $\mathbf{y}$. We also note that $p(c|\mathbf{x},\mathbf{z}) = p(c|\mathbf{z})$ because $c$ is a latent variable that only interacts with the latent representation of the input $\mathbf{x}$, i.e. $\mathbf{z}$. Therefore, conditional on $\mathbf{z}$, $c$ is independent of $\mathbf{x}$. $\qquad\square$

**Proposition 2.** *Choosing $q(c|\boldsymbol{z},\boldsymbol{y}))$ requires us to decompose $\mathcal{L}_{ELBO}$ as follows*

$$
\begin{aligned}
\mathcal{L}_{ELBO} &= \mathbb{E}_{q(z,c|x,y)} \log p(\boldsymbol{y}|\boldsymbol{z},c) + \mathbb{E}_{q(z|x)} \log \frac{p(\boldsymbol{x},\boldsymbol{z})}{q(\boldsymbol{z}|\boldsymbol{x})} \\
&\quad - \mathbb{E}_{q(z|x)}\mathbb{KL}\left(q(c|\boldsymbol{z},\boldsymbol{y})||p(c|\boldsymbol{z})\right).
\end{aligned}
\tag{11}
$$

*Proof.*

$$
\begin{aligned}
\boldsymbol{\mathcal{L}}_{\text{ELBO}} &= \mathbb{E}_{q(\mathbf{z},c|\mathbf{x},\mathbf{y})} \log p(\mathbf{y}|\mathbf{z},c) \\
&\quad + \mathbb{E}_{q(\mathbf{z},c|\mathbf{x},\mathbf{y})} \log \frac{p(\mathbf{x},\mathbf{z},c)}{q(\mathbf{z},c|\mathbf{x},\mathbf{y})} \\
&= \mathbb{E}_{q(\mathbf{z},c|\mathbf{x},\mathbf{y})} \log p(\mathbf{y}|\mathbf{z},c) \\
&\quad + \mathbb{E}_{q(\mathbf{z},c|\mathbf{x},\mathbf{y})} \log \frac{p(c|\mathbf{x},\mathbf{z})p(\mathbf{x},\mathbf{z})}{q(\mathbf{z}|\mathbf{x})q(c|\mathbf{z},\mathbf{y})} \\
&= \mathbb{E}_{q(\mathbf{z},c|\mathbf{x},\mathbf{y})} \log p(\mathbf{y}|\mathbf{z},c) \\
&\quad + \mathbb{E}_{q(\mathbf{z}|\mathbf{x})}\mathbb{E}_{q(c|\mathbf{z},\mathbf{y})} \left[ \log \frac{p(\mathbf{x},\mathbf{z})}{q(\mathbf{z}|\mathbf{x})} - \log \frac{q(c|\mathbf{z},\mathbf{y})}{p(c|\mathbf{z})} \right] \\
&= \mathbb{E}_{q(\mathbf{z},c|\mathbf{x},\mathbf{y})} \log p(\mathbf{y}|\mathbf{z},c) + \mathbb{E}_{q(\mathbf{z}|\mathbf{x})} \log \frac{p(\mathbf{x},\mathbf{z})}{q(\mathbf{z}|\mathbf{x})} \\
&\quad - \mathbb{E}_{q(\mathbf{z}|\mathbf{x})}\mathbb{KL}\left(q(c|\mathbf{z},\mathbf{y})||p(c|\mathbf{z})\right)
\end{aligned}
\tag{12}
$$

$\qquad\square$

**Proposition 3.** *The solution to the following convex program*

$$\min_{q(c|z,y)} \quad f_0(q) = \mathbb{KL}\left(q(c|\boldsymbol{z},\boldsymbol{y})||p(c|\boldsymbol{z})\right)$$

$$- \mathbb{E}_{q(c|\boldsymbol{z},\boldsymbol{y})} \log p(\boldsymbol{y}|\boldsymbol{z},c)\,, \tag{13}$$

$$s.t. \quad \sum_k q(c=k|\boldsymbol{z},\boldsymbol{y}) = 1, \quad q(c=k|\boldsymbol{z},\boldsymbol{y}) \geq 0, \quad \forall k$$

*is*

$$q(c=k|\boldsymbol{z},\boldsymbol{y}) = \frac{p(\boldsymbol{y}|\boldsymbol{z},c=k) \cdot p(c=k|\boldsymbol{z})}{\sum_k p(\boldsymbol{y}|\boldsymbol{z},c=k) \cdot p(c=k|\boldsymbol{z})}\,. \tag{14}$$

*Proof.* First, we note that the constraint, $q(c=k|\mathbf{z},\mathbf{y}) \geq 0$ for all $k$, is not needed (and effectively redundant), as the $\mathbb{KL}$ term in the objective function is not defined otherwise. Now consider a convex program that takes the form of

$$\min_{\mathbf{t}\in\mathbb{R}_+^k} \quad f_0(\mathbf{t})$$

$$s.t. \quad \mathbf{1}^T\mathbf{t} = 1\,. \tag{15}$$

where $f_0$ is a convex function and "$\succeq$" denotes "element-wise greater than or equal to." Forming the Lagrangian, we have

$$\mathbf{L}\left(\mathbf{t},\gamma\right) = f_0(\mathbf{t}) + \gamma\left(\mathbf{1}^T\mathbf{t} - 1\right)$$

The *Karush–Kuhn–Tucker conditions* state that the optimal solution dual, $(\mathbf{t}^*,\gamma^*)$, satisfies the following

- $-\mathbf{t}^* \preceq 0$

- $\mathbf{1}^T\mathbf{t}^* - 1 = 0$

- $\nabla_{\mathbf{t}}\mathbf{L}\left(\mathbf{t}^*,\gamma^*\right) = 0$

Since

$$\nabla_{\mathbf{t}}\mathbf{L}\left(\mathbf{t},\gamma\right) = \nabla_{\mathbf{t}}f_0(\mathbf{t}) + \gamma \cdot \mathbf{1}$$

the third condition implies that

$$\nabla_{\mathbf{t}}\mathbf{L}\left(\mathbf{t}^*,\gamma^*\right) = \nabla_{\mathbf{t}}f_0(\mathbf{t}^*) + \gamma^* \cdot \mathbf{1} = 0\,. \tag{16}$$

Let $\mathbf{t} = q(c|\mathbf{z},\mathbf{y})$ (i.e. $t_k = q(c=k|\mathbf{y},\mathbf{z})$), and $f_0(\mathbf{t})$ as being specified in Eq. 13, we have

$$\nabla_{t_k} f_0(\mathbf{t})$$

$$= \frac{\partial}{\partial t_k}\left[\sum_k t_k\left(\log \frac{t_k}{p(c=k|\mathbf{z})} - \log p(\mathbf{y}|\mathbf{z},c=k)\right)\right] \tag{17}$$

$$= \log \frac{t_k}{p(c=k|\mathbf{z})} + 1 - \log p(\mathbf{y}|\mathbf{z},c=k)\,.$$

Based on the condition in Eq. 16, we thus have

$$\nabla_{t_k}\mathbf{L}\left(\mathbf{t}^*,\gamma^*\right) = \log \frac{t_k^*}{p(c=k|\mathbf{z})} + 1$$

$$- \log p(\mathbf{y}|\mathbf{z},c=k) + \gamma^* \tag{18}$$

$$= 0$$

which leads to

$$t_k^* = e^{\log p(\mathbf{y}|\mathbf{z},c=k)-1-\gamma^*} \cdot p(c=k|\mathbf{z})\,.$$

Since $\gamma^*$ is chosen in a way such that $\sum_k t_k^* = 1$ (by the second condition), we obtain the solution

$$t_k^* = \frac{t_k^*}{\sum_k t_k^*} = \frac{p(\mathbf{y}|\mathbf{z}, c = k) \cdot p(c = k|\mathbf{z})}{\sum_k p(\mathbf{y}|\mathbf{z}, c = k) \cdot p(c = k|\mathbf{z})} \; . \tag{19}$$

$\square$

**Theorem 1.** *The variational lower bound for* `DGC` *is*

$$\log p(\boldsymbol{x}, \boldsymbol{y}) \geq \mathbb{E}_{q(\boldsymbol{z}, c|\boldsymbol{x}, \boldsymbol{y})} \log p(\boldsymbol{y}|\boldsymbol{z}, c)$$

$$+ \underbrace{\mathbb{E}_{q(\boldsymbol{z}, c|\boldsymbol{x}, \boldsymbol{y})} \log \frac{p(\boldsymbol{x}, \boldsymbol{z}, c)}{q(\boldsymbol{z}, c|\boldsymbol{x}, \boldsymbol{y})}}_{\text{ELBO for VAE with GMM prior}} \tag{20}$$

*Proof.* We derive the $\mathcal{L}_{\text{ELBO}}$ as follows

$$
\begin{aligned}
\log p(\mathbf{x}, \mathbf{y}) &= \log \int_{\mathbf{z}} \sum_c p(\mathbf{x}, \mathbf{y}, \mathbf{z}, c) d\mathbf{z} \\
&= \log \int_{\mathbf{z}} \sum_c \frac{p(\mathbf{x}, \mathbf{y}, \mathbf{z}, c)}{q(\mathbf{z}, c|\mathbf{x}, \mathbf{y})} q(\mathbf{z}, c|\mathbf{x}, \mathbf{y}) d\mathbf{z} \\
&\geq \mathbb{E}_{q(\mathbf{z}, c|\mathbf{x}, \mathbf{y})} \log \frac{p(\mathbf{x}, \mathbf{y}, \mathbf{z}, c)}{q(\mathbf{z}, c|\mathbf{x}, \mathbf{y})} \\
&= \underbrace{\mathbb{E}_{q(\mathbf{z}, c|\mathbf{x}, \mathbf{y})} \log p(\mathbf{y}|\mathbf{z}, c)}_{\text{Probabilistic Ensemble}}
\end{aligned}
\tag{21}
$$

$$+ \underbrace{\mathbb{E}_{q(\mathbf{z}, c|\mathbf{x}, \mathbf{y})} \log \frac{p(\mathbf{x}, \mathbf{z}, c)}{q(\mathbf{z}, c|\mathbf{x}, \mathbf{y})}}_{\text{ELBO for VAE with GMM prior}} \; .$$

$\square$

## B   Semi-supervised Clustering Methods

Semi-superivsed clustering methods usually assume the cluster labels of some observations are known, or certain observations are known to belong to the same cluster (e.g. the must-link and cannot-link constraints). `DGC` is different from traditional semi-supervised approaches in the following ways

- `DGC` does not assume access to true cluster labels for any observations. Specifically, the side-information is information that is *relevant* to the desired clusters (as shown in the experiments) but does not contain cluster labels on its own.

- The relation between the side-information and the desired clusters needs to be jointly *learned* with the clustering process and is *not* pre-given (compared to constraint clustering approaches where the constraints are assumed to be given). Moreover, the side-information `DGC` is able to incorporate, for instance the continuous side-information in Sec. 5.2, usually cannot be easily turned into constraints. Even for the binary side-information we used in Sec. 5.1 and Sec. 5.4, observations that belong to different clusters can still have the same side-information and vice versa. Therefore, without more prior knowledge, it is impossible to turn the binary side-information into constraints manually.

## C   Experimental Details

This section provides a detailed description of the experimental setups, such as the train/test splits, the chosen network architectures, and the choices of learning rate and optimizer, for the experiments conducted. We describe the architecture of `DGC` in terms of its encoder, decoder, and task network. We adopt the following abbreviations for some basic network layers

- $\text{FL}(d_i, d_o, f)$ denotes a fully-connected layer with $d_i$ input units, $d_o$ output units, and activation function $f$.

- $\text{Conv}(c_i, c_o, k_1, f, \text{BatchNorm2d}, O(k_2, s))$ denotes a convolution layer with $c_i$ input channels, $c_o$ output channels, kernel size $k_1$, activation function $f$, the 2D batch norm operation, and the spooling operation $O(k_2, s)$ with another kernel size $k_2$ and stride $s$.

## C.1 A Note on Feature Representations of Images

Following some previous works on incorporating deep learning into clustering (Jiang et al., 2017; Yang et al., 2019), we note that for images in the CIFAR 100-20 and SVHN datasets, instead of operating directly on the raw pixels, we use a pre-trained network to extract feature representations for images. It is worth pointing out that Jiang et al. (2017); Yang et al. (2019) used a pre-trained ResNet (He et al., 2016) as the feature extractor. This might raise some concerns regarding the nature of their methods being unsupervised, as the family of pre-trained ResNets are pre-trained on ImageNet (Russakovsky et al., 2015) which does utilize supervised information. To remedy this concern, we use DINO (Caron et al., 2021), a vision transformer trained in a self-supervised manner, as the feature extractor. We then use those feature representations as inputs to our framework.

## C.2 Pre-trained Networks for Side-information y

Recall that since at test time we do not assume access to the side-information $\mathbf{y}$, we pre-train a network using the input $\mathbf{x}$ to predict $\mathbf{y}$ so that we can use the predictions $\hat{\mathbf{y}}$ at test time. In this section we detail the architectures and performances of the pre-trained models for each experiment

- The pre-trained network achieves a classification accuracy of 98.7% for Noisy MNIST with the following architecture

| **Pre-trained Net for Noisy MNIST** |
|:---:|
| $\text{FL}(512, 1, \texttt{Sigmoid})$ |

- The pre-trained network achieves a MSE of 0.0048 for Pacman with the following architecture

| **Pre-trained Net for Pacman** |
|:---:|
| $\text{FL}(2, 20, \texttt{ReLU})$ |
| $\text{FL}(20, 1, \text{---})$ |

- The pre-trained network achieves a classification accuracy of 78.3% for CIFAR 100-20 with the following architecture

| **Pre-trained Net for CIFAR 100-20** |
|:---:|
| $\text{FL}(2048, 512, \texttt{ReLU})$ |
| $\text{FL}(512, 100, \texttt{SoftMax})$ |

- The pre-trained network achieves a classification accuracy of 73.8% for CBCS with the following architecture

| **Pre-trained Net for CBCS** |
|:---:|
| $\text{FL}(512, 1, \texttt{Sigmoid})$ |

- The pre-trained network achieves a classification accuracy of 82.3% for SVHN with the following architecture

| **Pre-trained Net for SVHN** |
|:---:|
| $\text{FL}(2048, 512, \texttt{ReLU})$ |
| $\text{FL}(512, 10, \texttt{SoftMax})$ |

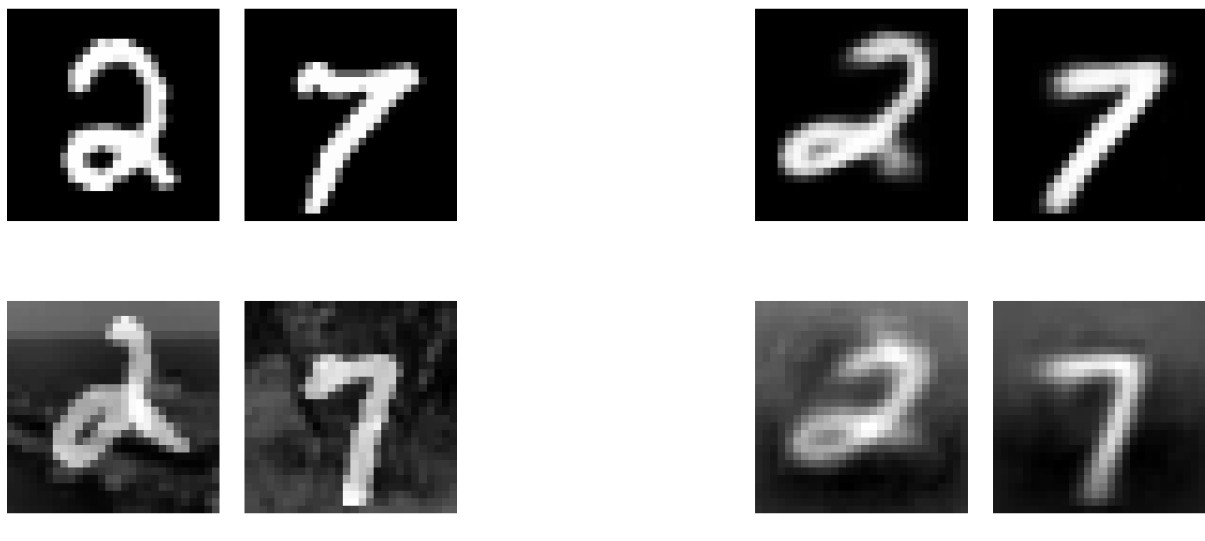

(a) Ground Truth Images             (b) Generated Images from DGC

Figure 5: Ground truth and generated noisy MNIST images.

## C.3  Noisy MNIST

We extract images that correspond to the digits 2 and 7 from MNIST. The MNIST dataset is pre-divided into training/testing sets, so we naturally use the images that correspond to the digits 2 and 7 from the training set as our training data (12,223 images), and that from the testing set as our testing data (2,060 images). For each digit, we randomly select half of the images for that digit and superpose noisy backgrounds onto those images, where the backgrounds are cropped from randomly selected CIFAR-10 images (more specifically, we first randomly select a class, and then randomly select a CIFAR image that corresponds to that class). See Fig. 1 for the ground truth and generated noisy MNIST samples.

We use the `Adam` optimizer for optimization. We train with a batch size of 128 images, an initial learning rate of 0.002, and a learning rate decay of 10% after every 10 epochs, for 100 epochs. We use the following network architecture:

| Encoder |
|---|
| FL(784,500,ReLU) |
| FL(500,500,ReLU) |
| FL(500,2000,ReLU) |
| FL(2000,10,—) |

| Decoder |
|---|
| FL(10, 2000,ReLU) |
| FL(200,500,ReLU) |
| FL(500,500,ReLU) |
| FL(500,784,ReLU) |

| Task Network |
|---|
| FL(10, 2,Sigmoid) |

## C.4  Pacman

This section provides more details for our Pacman experiments.

### C.4.1  Experimental Setup

We create 20,000 points, with 10,000 for the outer annulus and 10,000 for the inner annulus. Both annuli center at the origin, with the outer annulus having a radius of 1 and the inner annulus having a radius of 0.8. We create the training set by sampling 7,500 points from each annulus, and leave the rest of the data for

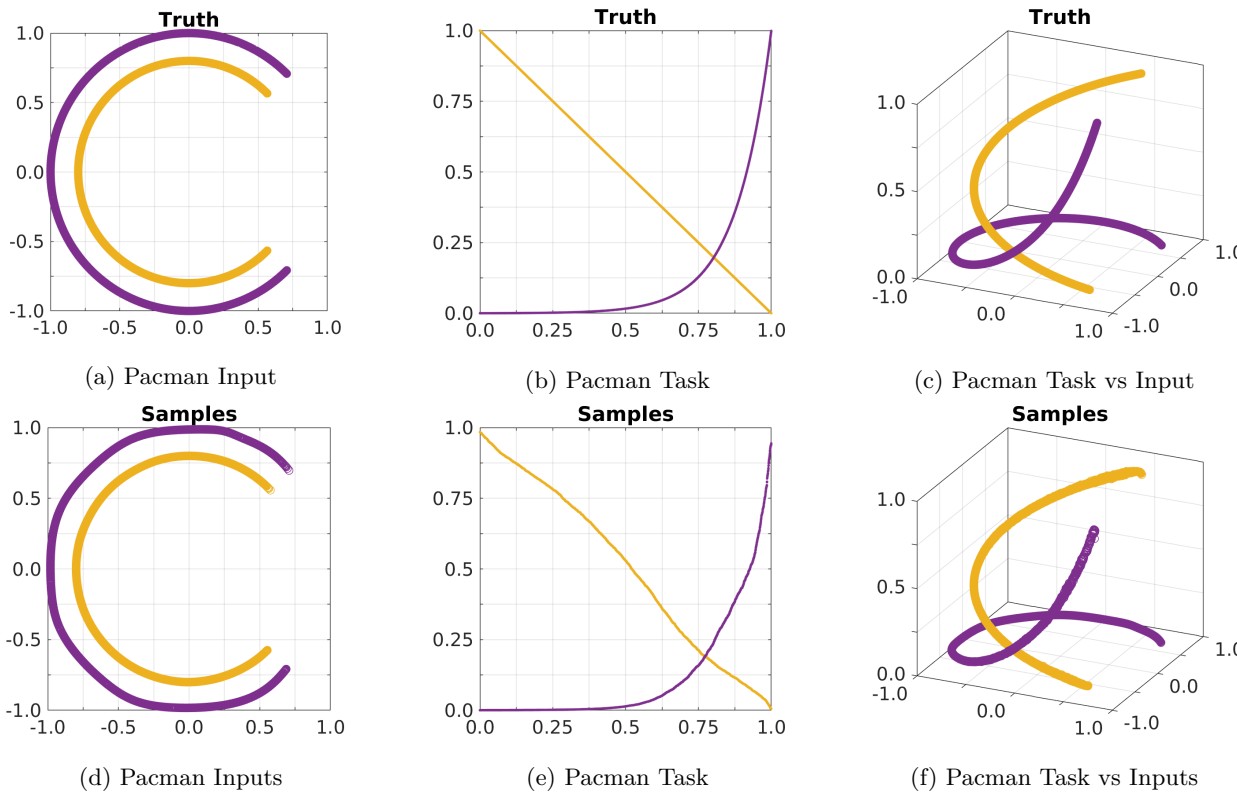

Figure 6: The first row shows the ground truth 2D Pacman, the responses **y** alone, and the combined 3D Pacman. The second row depicts the corresponding generated counterparts from `DGC`.

testing. We create the linear responses by dividing the [0,1] range into 10,000 sub-intervals and assign the split points to the points in the inner annulus in a way that it is increasing (from 1 to 0) in the clockwise direction. We create the exponential responses by evaluating the exponential function at the aforementioned split points (generated for the linear responses), and then assign them to the points on the outer annulus in a way that it is decreasing (from 0 to 1) in the clockwise direction. Fig. 6 shows the ground truth (first row) and the generated (second row) 2D Pacman annuli, the responses, and the 3D view of the entire dataset.

We use the `Adam` optimizer for optimization. We train with a batch size of 1,000 points, an initial learning rate of 0.001, and a learning rate decay of 10% after every 10 epochs, for 80 epochs. We use the following network architecture:

| Encoder |
| --- |
| FL(2,64,Sigmoid) |
| FL(64,128,Sigmoid) |
| FL(128,256,Sigmoid) |
| FL(256,60,—) |

| Decoder |
| --- |
| FL(60, 256,Sigmoid) |
| FL(256,128,Sigmoid) |
| FL(128,64,Sigmoid) |
| FL(64,2,Sigmoid) |

| Task Network |
| --- |
| FL(64, 128,Sigmoid) |
| FL(128,4,Sigmoid) |

## C.5 CIFAR 100-20

We apply `DGC` to the CIFAR 100-20 dataset. The training/test split of the dataset is given, with 60000 images in the training set and 10000 in the test set. As mentioned earlier, we apply `DGC` to the feature representations of the images rather than the images themselves.

We use the `Adam` optimizer for optimization. We train with a batch size of 256, an initial learning rate of 0.002, and a learning rate decay of 10 for every 10 epochs . We use the following network architecture:

| Encoder |
| --- |
| FL(2048,500,ReLU) |
| FL(500,500,ReLU) |
| FL(500,2000,ReLU) |
| FL(2000,10,—) |

| Decoder |
| --- |
| FL(10, 2000,ReLU) |
| FL(200,500,ReLU) |
| FL(500,500,ReLU) |
| FL(500,2048,ReLU) |

| Task Network |
| --- |
| FL(10, 2000, SoftMax) |

## C.6 Carolina Breast Cancer Study (CBCS)

### C.6.1 Data Processing

Due to the fact that the histopathological images collected in CBCS are large (of size $3 \times 3000 \times 3000$), we use a pretrained VGG16 network to extract feature representations for each image, and use the extracted, fixed features as the input to `DGC`. The features are of dimension 512, and are the output of the $8^{\text{th}}$ layer of the pretrained VGG16 network.

As mentioned in the main manuscript, each patient has 2-4 associated histopathologial images. Due to the scarce nature of medical data, we treat each image as an individual "patient" during training. At test time, we obtain patient-level prediction by aggregating image-level predictions (i.e. taking majority vote), and disregard patients with ambiguous patient-level predictions (e.g. a patient has 4 associated images. 2 of the images are predicted to be in cluster 0 and the other 2 are predicted to be in cluster 1). The number of disregarded patients accounts for 3.4% of the entire population.

Finally, again due to the scarce nature of this dataset, we use 10-fold cross validation to obtain predictions on the entire dataset. More specifically, we split the dataset into 10 subsets, train on 9 of those subsets and predict on the remaining subset. We then repeat this process 10 times to obtain the final predictions on the entire dataset.

### C.6.2 Experimental Setup

We use the `Adam` optimizer for optimization. We train with a batch size of 32, and an initial learning rate of 0.001 and decay rate of 0.9 (for every 10 epochs), for 150 epochs. The network architecture used is as follows

| Encoder |
|---|
| FL(512,1024,ReLU) |
| FL(1024,2048,ReLU) |
| FL(2048,5,—) |

| Decoder |
|---|
| FL(5, 2048,ReLU) |
| FL(2048,1024,ReLU) |
| FL(1024,512,ReLU) |

| Task Network |
|---|
| FL(5,3,Sigmoid) |

