# OpenReview forum: "Deep Goal-Oriented Clustering"
_TMLR — Rejected by TMLR_

### Review · Reviewer_FWTW · 2023-11-05

**Summary Of Contributions:**

The authors focus on the problem of clustering with side information. They structure their model as a variational autoencoder with Gaussian mixture model prior, thus learning a clustered representation for data in a manner that capture its density (unsupervised) but also being informative of the side-data (supervised).

**Audience:**

No

**Claims And Evidence:**

No

**Requested Changes:**

In general, the paper needs to be better organized with a more clear formulation and motivation for the assumptions made by the model and the relevance of the experiments in the context of real-world scenarios.

The related work section needs to be rewritten to have a better coverage of the existing literature, especially in relation to works that use autoencoders with supervision and autoencoders with structured priors.

The last line of (9) needs to be better justified as q(z|x) is used twice without explanation. Similarly, the p(c|x,z)=p(c|z) needs to be justified.

The authors should justify the practicality of the scenario like the one presented with artificial data in Section 5.2.

How is the cluster accuracy computed when the number of clusters is mis-specified?

What hypothesis test was used for the p-values in Section 5.4? How is RRD in Table 3 defined? what are the implications of 95% CIs crossing 0? Table 3 needs to be discussed in the text. Further, the quantities in tables 4 and 5 need a quantitative measurement of the association between the clusters and covariates (ER status, Grade and tumor subtype).

**Strengths And Weaknesses:**

The methodology proposed by the authors though in principle reasonable and addressing and addressing an important problem, has significant weaknesses. First, the formulation is overly complicated and confusing provided that it seems to be presented as a formal variational formulation yet some of the derivations lack details, are not well justified and in places it seem inconsistent with other parts of the formulation. Second, the experiments are for the most part underwhelming or not convincing because of the lack of comparisons with alternatives in the literature, artificial scenarios not motivated by real use cases and comparisons that are either unfair (comparing methods that do not have access to the same data) or qualitative (see below about Section 5.4). Additional details of the three points above are described below.

Provided the coverage of the related work section, which is very superficial, the authors do not seem aware of the substantial amount of existing work on supervised clustering (with autoencoder architectures or factor models) and autoencoders with mixture priors, including clustering with survival outcomes (relevant to the only real-world scenario considered in Section 5.4). It is also worth noting that the related work does not reference any publication after 2020 and that the baselines used in the experiments section are not discussed.

It is not clear why the authors write the variational posterior as a function of y to then say that they will ignore y as an input to the encoder. Further, the equality in (2) is not justified and z^(l) and M are not defined.

The authors claim that the standard VAE is a special case of DGC, however, it seems a somewhat moot statement provided that i) Kingma et al. (2014) does not consider labels (but there are quite a few models that do but are not mentioned), and more importantly, ii) DGC is not optimizing (1).

The motivation for making c a function of z (so z is a sufficient statistic for x) and y to then replace y for a pseudo-value predicted from a model is not clear because is not consistent with the generative process, but also because if in principle z is a sufficient statistic for x that can both recover x and predict y according to the generative model, why is there a need to make the variational posterior an explicit function of y to then replace it by an approximation.

The regularizer in (7) needs a better motivation and justification, specially when the side information is multivariate and/or continuous like in Section 5.2.

Section 5.1: The experiment is problematic for a few reasons: i) it is not fair to compare DGC with VaDE considering that the latter has access to information that is not available to the former, ii) no performance comparisons with other supervised clustering approaches are provided, iii) the experiment is not really about clustering, but classification, in which case it is not clear why classification baselines were not considered, iv) it is not clear why the model without the decoder will perform not better than random just by looking at the objective that is being optimized, so a proper justification is needed.

Section 5.2: The authors do not compare with other clustering, supervised clustering or semi-supervised approaches from the literature other than VaDE. Further, the generated data seems too contrived for a practical use case.

Section 5.3: The results in Table 2 are underwhelming because the baselines (without concat) do not seem to have access to the labels thus comparisons are not fair and no other method which can use labels (supervised or semi-supervised) are considered. It is not clear in which scenario one will have access to fine-grained labels but being interested in the (known) set of super-classes. Further, having access to the side information (i.e., the labels), why would one use them as covariates as in VeDE-Concat or SPICE-Concat as opposed to build a classifier or if labels are limited, building a semi-supervised model.

Section 5.4: The results are not convincing because: i) some details about Figure 4 and Table 3 are missing, ii) Tables 4 and 5 lack quantitative association comparisons, iii) there is no reasoning for the 3 cluster choice, iv) the proposed model is only compared against VaDE despite the being a number of models proposed to clustering with survival outcomes, one of which is referenced by the authors.

---

> ### Author Response · Authors · 2023-12-21
> **Reply to Reviewer FWTW**
>
> Thank you for your detailed and insightful feedback. We address your concerns and requested changes as follows:
>
> # Weakness
> 1. We updated the Related Work section in reflection of the works that other reviewers have pointed out, including works on clustering with survival outcomes and utilizing deep learning frameworks for constrained clustering.
> 2. a). We choose to not use $y$ as a part of the input to the encoder because, just as in a normal predictive setting where we want to derive information from the input $x$ to predict the output $y$, we want to train the encoder to efficiently extract information from the input $x$ that is sufficient for both reconstructing $x$ and predict $y$. This can be critical for downstream scenarios where efficient representations of $x$ are needed and the side-information labels $y$ are missing. b). The equality in (2) follows because $q(\textbf{z},c|\textbf{x},\textbf{y}) = q(c|\textbf{x},\textbf{y}, \textbf{z}) \cdot q(\textbf{z} |\textbf{x})$. The summation in the right-hand-side of (2) is then the expectation of $\log p(\textbf{y}|\textbf{z},c’)$ over $q(c|\textbf{x},\textbf{y}, \textbf{z})$. It is updated in the manuscript for additional clarity.
> 3. Could you please clarify what are you referring to by “DGC is not optimizing (1)”? We wish to note that the final loss DGC optimizes is a regularized version of (1), i.e. (1) with the regularization proposed on section 4.4. (see (7)). Therefore, (1) is the loss that we optimize over in DGC (with some regularization).
> 4. a). We want to re-emphasize that we replace $y$ with a pseudo-value predicted from the prediction network because we want to follow the typical supervised learning regimes where $y$ is usually not available at test time to make DGC as applicable as possible. As we note in the manuscript at the end of section 4.3, one should obviously use the ground-truth value $y$ at test time should it be available. b). We make the variational posterior distribution of $c$ a function of $y$ to make use of the task networks we trained during training. As you astutely point out, the learned $z$ carries information to both reconstruct $x$ and $y$; however, directly clustering samples based on the learned representations $z$ using $p(c|z)$ (i.e. the unsupervised part of the model without $y$) is equivalent to (because of the Gaussian assumptions) assigning $z$ to the mixture component whose mean is closest to $z$ in a $L_2$ sense (scaled by the standard deviations and assuming a uniform prior distribution on the cluster index $c$). There is no guarantee the information regarding reconstructing $x$ and predicting $y$ that is inherited in $z$ is separable in any way (in a $L_2$ sense), so we think directly incorporating $y$ through the learned task networks is necessary to make sure the information regarding $y$ is properly taken into account.
> 5. We added further clarifications in the main manuscript regarding the case when $y$ is continuous. Specifically, the introduced regularization formulation in 4.4 works for both discrete and continuous $y$. When $y$ is continuous, $p(y|\textbf{z}, c)$ denotes the likelihood of $y$ under the assumed probability distribution, i.e. $p(\cdot|\textbf{z}, c)$. Then this regularization encourages high likelihood of $y$ only under one cluster (as likelihood by definition is non-negative and a low entropy of normalized $p(y|\textbf{z}, c)$ for a given $y$ across the clusters $c$ translates to $y$ resulting in high likelihood under only one cluster), similarly to the case of discrete $y$.
>
> # Requested Changes
> 1. We updated the Related work section and provided further clarifications in places that are potentially unclear as you pointed out above.
> 2. We added further clarifications in the manuscript to clarify the notations in (9).
> 3. Our goal of the scenario presented in section 5.2 is to illustrate DGC’s ability to incorporate continuous side-information $y$, i.e. to use the task networks for regression-related tasks. We choose this simple, toy example setup as we think it would be easy and straightforward for the readers to recognize the effect of using the continuous side-information and to grasp the message we intend to convey: when the unsupervised signals (the two annulli) are difficult to separate apart, DGC is able to learn from the continuous side-information (the generated linear and exponential values). Along with other experiments we presented, we believe DGC is capable of utilizing continuous side-information/regression tasks to aid clustering in real-world scenarios.

---

> > ### Author Response · Authors · 2023-12-21
> > **Reply to Reviewer FWTW (Continued)**
> >
> > 4. We use the linear assignment algorithm to match the ground truth clustering labels to the predicted clustering labels. Linear assignment algorithm does not require the number of ground truth clusters to match that of the predicted ones, hence allowing us to mis-specify the number of clusters. Denoting the number of ground truth clusters as $k$ and assuming we specify we want to cluster the data into $m$ clusters (assume $m > k$), after running the linear assignment algorithm, we take the $k$ largest matching clusters (between the ground truth and the predicted ones) for calculating the clustering accuracy.
> > 5. The RRD stands for “Recurrence Risk Difference” in Tab. 3. They are risk differences for recurrence, calculated using the Kaplan-Meier estimator. The 95% CIs crossing 0 implies that negative recurrence differences are statistically possible; however, the vast majority of the CIs lie in the positive range for both cases. Moreover, we note that the CIs are wide partially because we used a relatively conservative non-parametric bootstrap for computing the CIs. The p-values are obtained from a log-rank test for the Kaplan-Meier model. In Tab. 4 and Tab. 5, the measures are the proportions of each stratum of the biological/clinical variables. We present that the proportions of the more indolent disease characteristics (ER+, lower grade, Luminal A status) are higher in cluster 0 relative to either cluster 1 or 2.

---

### Review · Reviewer_iB1Z · 2023-11-07

**Summary Of Contributions:**

I have already reviewed this paper 8 months ago and the authors did not incorporate the majority of my previous comments and suggestions. I therefore re-state them below:

***Summary:***
This work explores the integration of side information into the VaDE clustering algorithm to boost the clustering performance. It can be seen as an extension of the semi-supervised VAE by Kingma et al. to allow clustered latent embeddings. The authors showed the effect of adding side information on different applications.

**Audience:**

Yes

**Broader Impact Concerns:**

I believe there are no significant concerns about the ethical implications of the proposed approach.

**Claims And Evidence:**

No

**Requested Changes:**

1) I believe the main weakness is the lack of references and comparisons. I would strongly encourage the authors to add a thorough related work discussion.
2) Fix the imprecise claims.
3) Add the experiments described in my previous comment and add the code for reproducibility.
For additional details see above.

**Strengths And Weaknesses:**

***Strengths:***
The proposed approach provides a general framework to use both categorical and real-valued side information to drive the clustering algorithm toward a preferred configuration. I think the topic is of great interest to the community, however, the novelty and the performed experiments are still limited, and several previous works are ignored by the authors.

***Weaknesses:***

Limited novelty: The proposed generative approach is exactly the same as the one proposed in [1], resulting in the same ELBO. The only difference lies in p(yIz,c), while in [1] the latter is modeled as survival probability, the proposed approach uses standard regression/classification. I would strongly advise the authors to cite [1] and provide a thorough comparison in the paper.

Missing reference: The authors cite only a few works in the context of clustering with side information and exclude all works from 2019 on, however, many more recent approaches use pairwise constraints in deep clusterings, such as DC-GMM [5] and C-IDEC [3-4]. The authors cited Chapfuwa et al. and claimed that ‘the aforementioned approaches cannot be easily scaled to large, high-dimensional datasets’. First, this is a wrong statement, as Chapfuwa's paper can deal with high-dimensional datasets, secondly, there have been many more approaches that overcame this limitation (e.g. deep survival machines [2]).

Imprecise claims: I believe there are several statements that are not fully supported throughout the paper. As a concrete example, the authors state that (a) the side-information considered is ‘much more general’ than pairwise constraints, and (b) they ‘let the model learn what it deems useful from the side-information'. I would argue that pairwise information can be seen as weaker information than side-information as it does not provide any exact label, which the model needs to predict, but only a looser concept of similarity of data points. Additionally, pairwise constraints are usually collected for a very small proportion of samples while the side information used by the authors does not deal with missing labels.

Experiments: while the experiments tackle several problems, I think they do not fully support the authors' claim. In particular, in all sections the used side-information is indeed highly correlated with the clustering assignments, it would be interesting to see how the model performs if the side-information is loosely connected (but not uninformative) with the desired clustering. Additionally, in section 5.3. the setting of having fine-grained information is quite unrealistic and usually, one has the coarse-grained information and wishes to retrieve more fine-grained clusters [6].

Reproducibility: I could not find any available code, hence I am not able to check whether the proposed approach is reproducible.

[1] Manduchi et al. “A Deep Variational Approach to Clustering Survival Data.” ICRL (2021).

[2] Nagpal et al. Deep survival machines: Fully parametric survival regression and representation learning for censored data with competing risks. IEEE Journal of Biomedical and Health Informatics (2021)

[3] Zhang et al. A framework for deep constrained clustering algorithms and advances. In Joint European Conference on Machine Learning and Knowledge Discovery in Databases, 2019.

[4] Zhang et al. A framework for deep constrained clustering. Data Mining and Knowledge Discovery, 35(2):593–620, 2021.

[5] Manduchi, et al. “Deep Conditional Gaussian Mixture Model for Constrained Clustering.” Neural Information Processing Systems (2021).

[6] Ni, Jingchao, et al. “Superclass-Conditional Gaussian Mixture Model For Learning Fine-Grained Embeddings.” ICRL (2022).

***Improvements over the last submission:***
In my previous review, I argued that the model cannot choose to discard the side information if it is not informative for the cluster assignments, thus reducing the clustering performance. However, the authors performed a small ablation experiment where they chose uninformative side information and the performance of their method is comparable to VaDE. I believe this to be an interesting experiment, however, I would still argue that it very much depends on the dimensionality of the latent space, as some dimensions must be allocated to predict the uninformative y. I would suggest digging deeper and measuring the different losses in this setting (is the prediction loss decreasing or increasing?).

---

> ### Author Response · Authors · 2023-12-21
> **Reply to Reviewer iB1Z**
>
> Thank you for your detailed and insightful feedback again. We address your concerns and requested changes as follows:
>
> # Weakness
> 1. Thank you again for bringing [1] to our attention. We agree that the generative process of DGC is similar to that proposed in [1], where, as you point out, we use standard classification/regression for the side-information whereas they focus on survival modeling. We, however, want to point out one of the biggest differences between DGC and the model proposed in [1]: how $q(c|\textbf{y},\text{z})$ (or in their case $q(c|\textbf{t},\text{z})$) is computed. In [1], they simply choose $q(c|\textbf{t},\text{z})$ to be the unsupervised probability $p(c|\textbf{z}, \textbf{t})$, the same as what was being done in the original unsupervised VaDE model (see [2]). As shown in [2], simply choosing the variation probability $q(c|\textbf{x})$ in VaDE to be $p(c|\textbf{x})$ maximizes the ELBO. However, as what we have shown, the same choice, as it was being done in [1], is sub-optimal in the presence of side-information. In comparison, in this work we analytically derived the optimal solution for $q(c|\textbf{y},\text{z})$ in terms of maximizing the ELBO (see section 4.3, specifically Eq.(5) and Eq.(6)). We showed in section 5.2, specifically Tab.1, that it is difficult for neural networks to recover this analytically derived optimal solution.
> 2. Thank you for your suggestions. We updated our related work section in reflection of the references you listed here.
> 3. Thank you for pointing out the difference and how realistic it is to have fine-grained information versus coarse-grained information. We chose to use the fine-grained classes as the side-information for clustering the coarse-grained information mainly due to the fact that most methods in the clustering literature only consider clustering the coarse-grained information for CIFAR 100-20. We indeed agree with you, and have included an experiment where we use the coarse-grained classes as the side-information for clustering the fine-grained labels (see Modeling Flexibility in section 5.3).
> 4. The code is now available at: https://github.com/uncbiag/dgc.
>
> # Improvement over the last submission
> Thank you for suggesting this experiment and we are glad you find this experiment interesting. We want to emphasize that when we compare DGC (with uninformative side-information $y$) with VaDE, the latent dimensions for both models are set equally. This is to say, although some dimensions are potentially allocated to predict the uninformative $y$ for DGC, the resulting clustering accuracies are still comparable. Moreover, we want to point out that the latent dimension used in the CIFAR 100-20 experiment (as stated in the Appendix) is 10, which is already relatively small and potentially indicates the effect of latent dimension on the performance is not dramatic.
>
> # Requested Changes
> 1. Thank you for pointing out the related works here, we have incorporated all the listed works here and updated our Related Work section.
> 2. We rephrased the claims to emphasize the fact that the goal of DGC is to utilize side-information (i.e. information that is related to the clustering task but not directly indicative of the final clustering labels) that is not in the form of explicit constraints.
> 3. We provided the code as stated above.
>
> [1] Manduchi et al. “A Deep Variational Approach to Clustering Survival Data.” ICRL (2021).
>
> [2] Jiang et al. “Variational Deep Embedding: An Unsupervised and Generative Approach to Clustering.” IJCAI, 2017.

---

### Review · Reviewer_tnSU · 2023-12-09

**Summary Of Contributions:**

The authors propose a latent variable model for data that exploits supervision of side information for unsupervised learning.  Specifically, they propose to incorporate what they term *indirect* side-information as a proxy for sharpening the contours of a latent variable density model in the form of a VAE, which they name Deep Goal-Oriented Clustering.  The formulation of indirect side-information differs from traditional side-information used in semi-supervised learning (typically consisting of cluster inclusion or exclusion data point constraints), comprising either discrete or continuous observables associated with the data $x$.  The authors derive a VAE inspired by prior work and propose to use a Gaussian mixture prior as well as a regularized ELBO in their framework.  They evaluate their model on different experiments, each designed to probe (or validate) a specific design choice.

**Audience:**

Yes

**Claims And Evidence:**

No

**Requested Changes:**

### Introduction

> However, with the explosion of the size of modern datasets, it becomes increasingly unrealistic to manually annotate all available data for training. Therefore, understanding inherent data structure through unsupervised clustering is of increasing importance.

- Maybe a citation to LeCun here?  Or cost of annotation lit?

### Related work

> If there exists a set of semantic labels associated with the data (e.g. the digit information for MNIST images), the cluster assumption states that there exists a direct correspondence between the labels and the clusters

- I’m not sure that Chapelle et al 2006’s statement of the cluster assumption agrees exactly with this; It is stated rather that there is a relationship between optimal decision boundaries and the density of data points:

“The cluster assumption states that the decision boundary should not cross high density regions, but instead lie in low density regions.”

The statement here is more rigid and direct; I would suggest reusing Chapelle et al’s phrasing, as it’s more general.

- The beginning of the subsection on joint modeling is awkward; perhaps the first sentence could define the term (I presume it refers to modeling joint probability distibutions, sometimes incorporating occasional observations of otherwise latent variables?)

### Deep Goal-Oriented Clustering

- In Figure 1, the arrows (espcially dashed) should be explained in the caption to the PGM graph.
- A nitpick in section 4.2, the factorization of the variational posterior q of $q(Z,c | X,Y)$ would be better off as $q(Z|X,Y) * q(c|Z,X,Y)$, so the variables agree in order of presentation in the joint distributiton.  Yes, they are equivalent.  But this makes it easier to read, and shows that the joint is factorized as a product of conditionals.

- Later on in 4.2, the authors write
> The probabilistic ensemble allows the model to maintain necessary uncertainty until an unambiguous clustering structure is captured.

This is a bit unclear; do the authors imply that, as the decoder converges on a more optimal solution for generating observations, the responsibilities term $\lambda_{c’}$ will sharpen and the weights will become sparse?  This seems like it might be possible on sanitized datasets like CIFAR or MNIST, but not in real data.


- Further on in 4.3, when discussing the $\lambda_{c}$ term:
> Analyzing the last term in Eq. equation 3, we notice that if the learned variational posterior q(c|x, z, y) is very discriminative and puts most of its weight on one specific index c, all but one KL terms in the weighted sum will be close to zero

Belabouring my previous point, it seems that the suitability of the choice to parameterize the encoder as a unimodal distribution is dependent on the assumption that the model can learn to separate the data by class, and that the $\lambda_{c’}$ will become sparse.  It’s far from clear that this will become so in cases where (e.g) the dataset has high aleatoric uncertainty.

- For cases where $y$ is continuous, how well does the predictor described at the end of section 4.3 work?  Would it be better to use a simpler model rather than an MLP?

- In section 4.4, the terminology used to describe entropy is a bit confusing in a few different ways.   I’ve never seen entropy referred to as an operator before.  Maybe use ‘function’ instead? Also, the authors  do not actually regularize the entropy, they use the entropy as a regularization term for the ELBO.
- Does the formulation of the regularizer with a softmax not imply that the side-information $y$ is assumed to be discrete?  I am a bit confused here about how this would work if $y$ is in fact continuous?  Would you atttempt to quantize $y$?

### Experiments

- In section 5.1:
> Instead, DGC is able to work with side-information that is only partially indicative of what the final clustering strategy should be, making DGC more applicable to general settings.

I don’t see how the authors can arrive at this conclusion looking only at the results in Figure 2.  Yes, the class label side information resulted in nearly eliminating all 2B,7B errors.  But I expected this given the design of the dataset; the class label information is independent of the background information.  I would be more convinced if there were some dependence of background inclusion based on the class label.  It’s also curious that, looking solely at the 2 vs 2B, VaDE makes no mistakes while DGC introduces two in 2 vs 2B.  Was this an unlucky artifact, or did the side information result in a misclassification?

- In section 5.2, the pacman problem seems more of a manifold learning problem; would it not be better to include LLE, MDS, or Laplacian Eigenmaps as comparators?
- For Table 1, I do not think it is really appropriate to call the class label $y$ side-information in the pacman task; it is exactly what the model is being evaluated upon to predict.  This is why VaDE achieves essentially random performance.  Unless there is something else meant by ‘separate the two annuli’ beyond ‘classify the points along each annulus’, I am not sure what conclusions to draw here.

- In section 5.3, specifically the paragraph at the end about **Jointly train the prediction network for the side information**; this is an interesting experiment, but only if the accuracy of the prediction network is shown; the results suggest that the prediction network his extremely accurate (meaning that mutual information between x,y is very high), but what about in cases where this isn’t the case?

**Strengths And Weaknesses:**

## Strengths

This paper takes a somewhat unusual perspective, asking what might be possible if we ignore the barrier between supervised and unsupervised learning.  They try to define their own niche apart from semi-supervised learning by considering side-information as either categorical labels or continuous descriptors.

- The writing is quite clear; the model is clearly described in section 4, with the extensions from classic VAEs made clear.

## Weaknesses

The principal weakness is a lack of precision when describing what information is being provided to each model in each experiment, as well as what the precise metric is, and how model predictions are produced.

For example in section 5.3, the authors state the goal is to cluster the images into the 20 super classes.  But given the notion of 20 super classes, this is a classification problem.  A clustering problem does not usually admit the number of clusters a priori.  It is also unusual to see accuracy used as a metric for clustering, given that clustering methods do not have any notion of class label. I think to help the reader, the authors should outline precisely how, given a clustering solution from each method, they calculate the accuracy reported.

Also in section 5.1, what is the measure ‘clustering accuracy’?  Is it homogeneity within clusters?  This is unusual usage of the term, which is typically used for supervised learning.

---

> ### Author Response · Authors · 2023-12-21
> **Reply to Reviewer tnSU**
>
> Thank you for your detailed and insightful feedback. We address your concerns and requested changes as follows:
>
> # Weakness
> ## The usage of ``clustering accuracy” as a metric
> Indeed “accuracy” is a widely-adopted metric in supervised learning, specifically in classification tasks. Nonetheless, “clustering accuracy” is popular metric in clustering literature as well (see [1] and [2] as an example). When use “clustering accuracy” as a metric, we measure the degree of agreeableness between the set of predicted cluster labels and the set of predefined ground truth labels (using linear assignment algorithms as the predicted cluster IDs usually do not align with the predefined labels semantically). However, we do agree that accuracy, by no means, is the best metric for clustering tasks. Automatically learning the number of clusters needed (through, for instance, nonparametric Bayesian methods), as you point out, is also highly desirable and is a future research direction we are pursing.
>
> # Requested Changes
> 1. We added a citation to the original ImageNet paper in which the manual labeling constituted the majority of the work to create this dataset.
> 2. We changed the phrasing on the “cluster assumption” based on your recommendation, clarified the beginning of the subsection on joint modeling, added more explanation to the caption of Fig.1, and rearranged the factorization in section 4.2.
> 3. We appreciate your insightful comments on the suitability of the choice to parameterize the encoder as a unimodal distribution. In reality, for the real breast cancer data we tested on in 5.4, we still observed that the final learned $\lambda_{c’}$ is very sparse (especially with the introduced regularization described in 4.4). Nevertheless, as you astutely point out, the suitability of this parametrization choice ultimately depends on the sparsity of $\lambda_{c’}$.
> 4. The pre-trained side-information predictor performs well in the case of continuous $y$ in the Pacman experiment. In a simple, toy example setting like this, a simpler model (like a regression model) might suffice. However, in a more complicated setting (like the CIFAR 100-20 where we essentially are training a CIFAR classification model, a neural network is likely needed).
> 5. We use the term “entropy operator” to be in line with the mathematical definition of a functional operator (see [3]), i.e. the inputs to this operator are functions (as $p(y|\textbf{z}, c)$ are probability distributions over $y$ that are parameterized as functions of $\textbf{z}$ and $c$). We further clarified that we add the entropy term to the ELBO as a regularization term.
> 6. The introduced regularization formulation in 4.4 works for both discrete and continuous $y$. When $y$ is continuous, $p(y|\textbf{z}, c)$ denotes the likelihood of $y$ under the assumed probability distribution, i.e. $p(\cdot|\textbf{z}, c)$. Then this regularization encourages high likelihood of $y$ only under one cluster (as likelihood by definition is non-negative and a low entropy of normalized $p(y|\textbf{z}, c)$ for a given $y$ across the clusters $c$ translates to $y$ resulting in high likelihood under only one cluster), similarly to the case of discrete $y$.
> 7. Regarding the induced errors from DGC in 2 vs 2B, we believe it’s an artifact as re-running the model with a different network initialization results in only 1 error (the network mis-clustered a sample of digit 2 as 2B (i.e. with background).
> 8. To the best of our knowledge, LLE, MDS, or Laplacian Eigenmaps are dimensionality reduction methods as opposed to clustering methods. The goal of this experiment is to demonstrate DGC’s ability to utilize continuous side-information $y$ to identify the two annuli whereas the unsupervised version of DGC, VaDE, is not able to do the same in an unsupervised manner.
> 9. We want to clarify that the side-information $y$ used in Pacman are the continuous values we generated. To quote from the manuscript:” These values are constructed such that they decrease linearly (from 1 to 0) in one direction for the inner (yellow) annulus, and increase exponentially (from 0 to 1) in the opposite direction for the outer (purple) annulus (see Figure 3a for a 3D illustration of the dataset).” They are \textbf{not} the class labels. DGC in the entire process never has access to the class labels, hence never “classifying the points along each annulus” but rather clustering the points with the help of the continuous side-information $y$. Our goal is to show by utilizing this continuous side-information, DGC is able to identify the two annuli, as demonstrated in Table 1.

---

> > ### Author Response · Authors · 2023-12-21
> > **Reply to Reviewer tnSU (Continued)**
> >
> > 10. The prediction accuracy of the prediction network is 83.2% when jointly trained with DGC. We find that as long as the side-information prediction accuracy is reasonable, DGC is not particularly sensitive to how extremely accurate the prediction network is (e.g. in this case we find a prediction accuracy of 83.2% suffices). We also conducted another experiment in 5.3 titled “Non-informative side-information” where we randomly generate $y$. As expected, we find that when the mutual information between $x$ and $y$ is low (or $y$ is non-informative), DGC resorts back to its unsupervised counterpart VaDE.
> >
> >
> > [1] Gansbeke et al. “SCAN: Learning to Classify Images without Labels.” ECCV (2020).
> >
> > [2] Niu et al. “SPICE: Semantic Pseudo-Labeling for Image Clustering.”  IEEE Transactions on Image Professing (2021).
> >
> > [3] https://en.wikipedia.org/wiki/Operator_(mathematics)

---

### Decision · Action_Editor_kkWD · 2024-01-15

**Recommendation:** Reject

**Comment:**

As discussed above, the reviewers are critical of how this work is situated within the related work, the motivation of the design choices, and the empirical results. Therefore, they currently do not feel comfortable accepting this paper and would suggest a major revision and resubmission.

**Audience:**

The topic of the paper would be interesting for at least some part of the TMLR audience. If the paper would be sufficiently improved following the reviewer's feedback, a resubmission should therefore be possible.

**Claims And Evidence:**

The reviewers feel that the presented work is not sufficiently situated within and empirically compared to the large amount of related work in this area, such that it is not clear to the reader in which ways it differs from other approaches. Moreover, many of the design decisions seem to strike the reviewers as questionable and not properly motivated. Lastly, the empirical results are perceived as weak.

**Resubmission Of Major Revision:**

The authors may consider submitting a major revision at a later time.